# Structural dynamics of protein-protein association involved in the light-induced transition of *Avena sativa* LOV2 protein

Changin Kim [1,2,4], So Ri Yun [1,2,4], Sang Jin Lee [1,2,4], Seong Ok Kim [1,2,4], Hyosub Lee [1,2], Jungkweon Choi [1,2], Jong Goo Kim [1,2], Tae Wu Kim [1,2], Seyoung You [1,2], Irina Kosheleva [3], Taeyoon Noh[1,2], Jonghoon Baek[1,2] & Hyotcherl Ihee [1,2] ✉

The Light-oxygen-voltage-sensing domain (LOV) superfamily, found in enzymes and signal transduction proteins, plays a crucial role in converting light signals into structural signals, mediating various biological mechanisms. While time-resolved spectroscopic studies have revealed the dynamics of the LOV-domain chromophore's electronic structures, understanding the structural changes in the protein moiety, particularly regarding light-induced dimerization, remains challenging. Here, we utilize time-resolved X-ray liquidography to capture the light-induced dimerization of *Avena sativa* LOV2. Our analysis unveils that dimerization occurs within milliseconds after the unfolding of the A'α and Jα helices in the microsecond time range. Notably, our findings suggest that protein-protein interactions (PPIs) among the β-scaffolds, mediated by helix unfolding, play a key role in dimerization. In this work, we offer structural insights into the dimerization of LOV2 proteins following structural changes in the A'α and Jα helices, as well as mechanistic insights into the protein-protein association process driven by PPIs.

Photoreceptors detect light signals, a critical environmental stimulus for living organisms, and convert them into biological signals[1,2]. This enables the organisms to adapt to and respond to changes in their light environment[2]. The LOV superfamily acts as photosensory units in various enzymes and signal transduction proteins with flavin mononucleotide (FMN) as their chromophores[1,3–5]. The core structure of LOV domain consists of five central β-sheets (Aβ, Bβ, Gβ, Hβ, and Iβ) and four α-helices (Cα, Dα, Eα, and Fα), which are highly conserved throughout blue-light receptors[1,6] (Fig. 1). Upon irradiation, LOV domains utilize flavin mononucleotide (FMN) chromophores to sense blue light, converting such stimuli into structural signals through conformational changes in the protein moiety[1,6–8].

Various studies have been conducted on the photoresponse of these LOV domains[1,9]. These studies have proposed that, upon activation by blue light, a photoadduct is formed through a covalent bond between FMN and cysteine located within the Eα of these domains, followed by subsequent structural changes in the protein moiety of LOV domains[10–13]. Especially, diverse photoreceptors that possess LOV domains as light-sensing regions undergo dimerization during their light-induced signal transduction[14–22]. These structural changes play a crucial role in modulating downstream biological processes.

Although spectroscopy studies have suggested that various LOV domains may form transient dimers[12–14,16–22], the dimerization observed in these studies was dependent on concentration, even in the absence of light[14,19,23–28]. Moreover, while ultrafast studies effectively probe the

---

[1]Department of Chemistry, Korea Advanced Institute of Science and Technology (KAIST), Daejeon 34141, Republic of Korea. [2]Center for Advanced Reaction Dynamics, Institute for Basic Science (IBS), Daejeon 34141, Republic of Korea. [3]Center for Advanced Radiation Sources, The University of Chicago, Chicago, IL 60637, USA. [4]These authors contributed equally: Changin Kim, So Ri Yun, Sang Jin Lee, Seong Ok Kim. ✉e-mail: hyotcherl.ihee@kaist.ac.kr

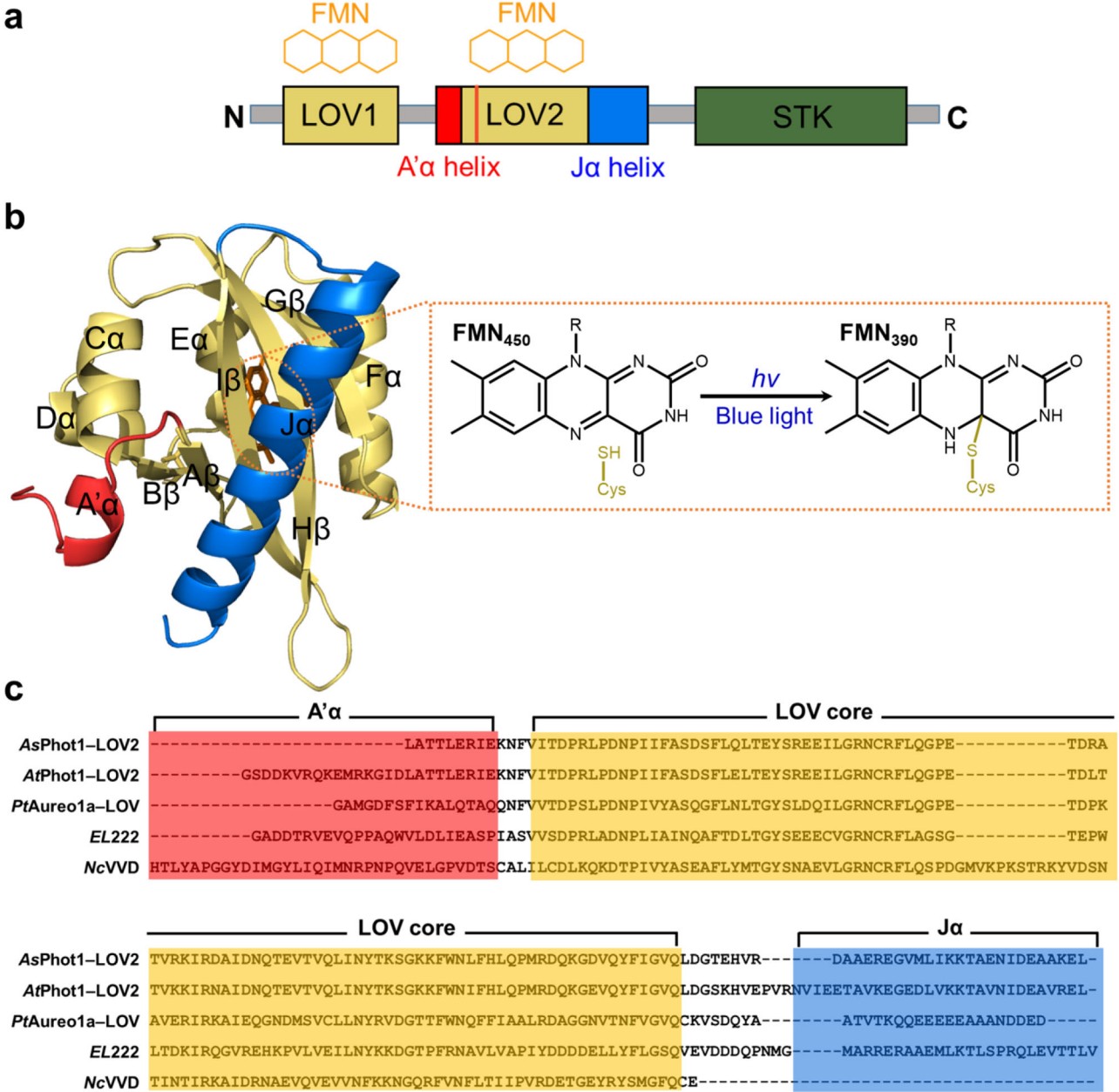

**Fig. 1 | Structure and photocycle of *As*LOV2. a** Domain architecture of *Avena sativa* phototropin 1 (*As*Phot1). *As*Phot1 consists of three functional domains, which are organized into LOV1 (yellow), LOV2 (yellow), and STK (green). LOV2 is flanked by A'α (red) and Jα (blue) helices. **b** The three-dimensional structure of *As*Phot1–LOV2 (*As*LOV2, A'α + LOV2 domain + Jα) and the light-induced photo-adduct (Cys–FMN) formation of FMN in *As*LOV2. In this experiment, we used an *As*LOV2 construct where the A'α (red) and Jα (blue) helices are coupled to the LOV2 domain (yellow). Upon blue-light irradiation, the photoreaction is initiated by the formation of a photoadduct (orange square dashed box) between the FMN chromophore and a cysteine residue. **c** Multiple sequence alignment of the LOV domains: *As*Phot1–LOV2 (*As*LOV2), *Arabidopsis thaliana* phototropin 1 LOV2 (*At*Phot1–LOV2, *At*LOV2), *Phaeodactylum tricornutum* Aureochrome 1a LOV (*Pt*Aureo1a–LOV), *Erythrobacter litoralis* 222 (*EL*222), and *Neurospora crassa* VVD (*Nc*VVD). The sequences of each species' LOV domain corresponding to A'α (red), Jα (blue) helices and the LOV core domain (yellow) are highlighted in colored boxes.

electronic structure of the chromophore, they do not directly detect global structural changes of the proteins involved in photo-induced dimerization. Notably, as summarized in Supplementary Fig. 1, several structural studies on the proteins have reported the dimer con-formations of LOV domains, suggesting that the proteins can form dimer conformations with various dimeric interactions[24,27–33]. How-ever, they do not offer detailed structural insights into the photo-induced dimerization. These limitations have made it challenging to obtain specific structural information about photo-induced dimeriza-tion, leaving the structural characteristics of LOV domain dimers and their dimerization process elusive.

In this work, to overcome these challenges, we apply time-resolved X-ray liquidography (TRXL) to capture the photoinduced structural dynamics in *As*LOV2, which does not show concentration dependence on dimerization in the dark state (Fig. 1 and Supple-mentary Fig. 2a, b). *As*LOV2 (residue 413-520) belongs to the LOV2 domain of *Avena sativa* phototropin 1 (*As*Phot1), sharing the same structural framework as canonical LOV domains[34]. We used a con-struct that incorporates the A'α helix (residue 404-412) connecting to the LOV1 domain on one side and the Jα helix (residue 521-546), which links to the serine/threonine kinase (STK) domain on the other side of *As*LOV2[34]. Due to the presence of the A'α and Jα helices, this

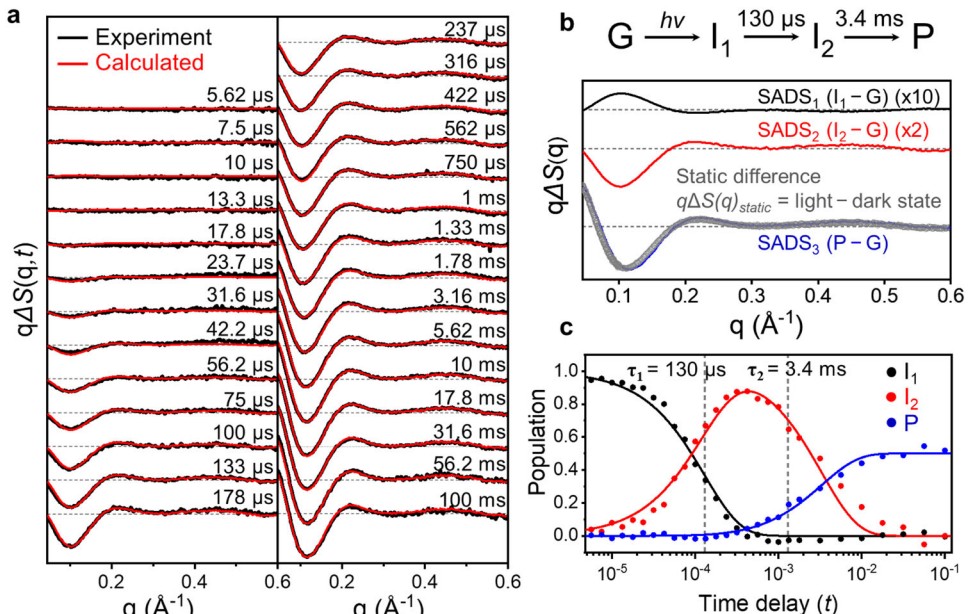

**Fig. 2 | Global kinetic analysis of the TRXL data of *As*LOV2 I427V. a** Experimental (black) and calculated (red) difference scattering curves. **b** The top panel represents the sequential kinetic model of *As*LOV2 determined in this study. The lower panel shows the SADSs obtained through KCA analysis. SADS₁ (black), SADS₂ (red), and SADS₃ (blue) correspond to I₁, I₂, and P (photoproduct) of the sequential kinetic model, respectively. The static difference curve (q$\Delta S_{static}$, light state – dark state, gray), normalized to SADS₃, is also shown. **c** Time-resolved changes in the population of transient species (I₁ (black), I₂ (red), and P (blue)). Since the protein exhibits dimer conformation in the P state, the maximum population of P is 0.5. This value is half those of the other states (G, I₁, and I₂), which exhibit monomer conformations.

construct maintains a monomeric state in the dark, making it a suitable model system for observing transient dimerization upon irradiation. In this work, we refer to this construct including the A'α and Jα helices (residue 404-546) as *As*LOV2. TRXL, also known as time-resolved X-ray solution scattering (TRXSS)[15,35–47], is one of the sensitive experimental methods for detecting global conformational changes in solution induced by a trigger such as optical excitation and temperature jump. When TRXL is applied to macromolecules, it is also referred to as TR small-angle/wide-angle X-ray scattering (TR-SAXS/TR-WAXS). Kinetic analysis of the TRXL data from both wild type (WT) and I427V mutant[48], the latter of which exhibits a relatively faster dark recovery compared to WT, identified three distinct intermediates connected by two time constants on the microseconds and milliseconds timescales in the light-induced structural transition of LOV2. Additionally, structural analysis aided by molecular dynamics (MD) simulations revealed that LOV2 undergoes extensive conformational changes related to the changes in oligomeric state (dimerization) through two monomeric intermediate states. Regarding the dimerization, we demonstrated that the formation of the dimeric interface for *As*LOV2 is associated with the inter-monomeric interaction between the β-scaffolds, as evidenced by size exclusion chromatography (SEC) experiments on three *As*LOV2 constructs: a construct with the Jα helix deleted (ΔJα), a construct with the A'α helix deleted (ΔA'α), and a construct with both Jα and A'α helices deleted (ΔJα/ΔA'α). Moreover, we observed that the SEC profiles of K413E and E475K mutants, designed to disrupt the electrostatic interaction between *As*LOV2 monomers, exhibit a decreased dimer fraction compared to that of the WT. This proposes that the PPIs, including not only the interaction between the β-scaffolds but also a local electrostatic interaction, contribute to the formation of the dimeric interface. The findings of our study provide direct insights into the structural mechanism of photo-induced dimerization in *As*LOV2, as well as a structural template for understanding the process of protein-protein association driven by PPIs[49,50].

## Results
### TRXL data and kinetic analysis

To investigate the light-induced structural changes of *As*LOV2, we measured TRXL data from microseconds (μs) to hundreds of milliseconds (ms) using the laser pump-X-ray probe scheme[36–39] (details in Methods). TRXL experiments were conducted for both WT and I427V mutant. The I427V mutant[48], which exhibits a faster photocycle than WT, was deliberately chosen to efficiently collect TRXL data with a higher signal-to-noise ratio (SNR), as the slow recovery of WT required the lowering of the repetition rate for measuring data at long time delays. The TRXL data were collected from 5.62 μs to 316 ms for WT and from 5.62 μs to 100 ms for the I427V mutant (Supplementary Fig. 3). The difference scattering curves, $\Delta S(\mathbf{q}, t)$, of the I427V mutant, show a positive peak with marginal scattering intensities in the small-angle region ($q < 0.2$ Å$^{-1}$) on the timescale from several μs to tens of μs (Fig. 2a). Within the timescale from tens of μs to hundreds of ms, the difference curves show the formation and increase of a significant negative peak in the small-angle region (Fig. 2a). During this relatively late timescale, the oscillatory features are also observed in the wide-angle region ($0.2 < \mathbf{q} < 0.6$ Å$^{-1}$) (Fig. 2a). To extract kinetic and structural information from the difference scattering curve, we conducted a global kinetic analysis using singular value decomposition (SVD) and kinetics-constrained analysis (KCA) methods for $\Delta S(\mathbf{q}, t)$ in the $\mathbf{q}$ range of 0.03 to 1.0 Å$^{-1}$ by following a well-established data analysis protocol (details in Methods)[39]. In SVD, the experimental data set of I427V was decomposed into time-independent $\mathbf{q}$-spectra (left singular vectors, LSVs), the weight of the singular vector (singular values, S), and time-dependent amplitude changes of the RSVs (right singular vectors, RSVs).

From SVD analysis, the three structurally different components were identified. The two different time constants of $130 \pm 16$ μs and $3.4 \pm 0.7$ ms were obtained by fitting the first three RSVs with the common time constants (Supplementary Figs. 4 and 5). In the subsequent kinetic analysis using KCA, we considered two kinetic models ((i) a sequential kinetic model and (ii) a parallel kinetic model)

including three transient components and related two-time constants of 130 µs and 3.4 ms obtained from the SVD analysis, and extracted species-associated difference scattering curves (SADSs) containing direct structural information of the transient species (Fig. 2 and Supplementary Fig. 6). The theoretical time-resolved difference scattering curves for each kinetic model were generated by a linear combination of SADSs. The theoretical difference curves for the sequential kinetic model show good agreement with experimental data in both small- and wide-angle regions, while those for the parallel kinetic model show notable discrepancies from the experimental data in the small-angle region (Supplementary Fig. 6). This suggests that the sequential kinetic model is more suitable for describing the light-induced structural dynamics of $As$LOV2 (Fig. 2). In the sequential kinetic model, the first intermediate ($I_1$) is formed within 5.62 µs, which is the first time delay of our TRXL measurement. Subsequently, the second intermediate ($I_2$), formed with a time constant of 130 µs is converted into the third intermediate ($I_3$) with a time constant of 3.4 ms (details of SVD and KCA analysis are described in Methods).

Among the three SADSs extracted from the sequential kinetic model (Fig. 2b), the first SADS ($SADS_1$), corresponding to $I_1$, shows a positive peak in the small-angle region. The scattering intensity of $SADS_1$ is marginal compared to the other two SADSs, indicating that the formation of $I_1$ involves relatively minor structural changes compared to the formation of the other species. In contrast, the second SADS ($SADS_2$), corresponding to $I_2$, exhibits a negative peak in the small-angle region, and its intensity is more than five times larger than that of $SADS_1$, indicating that the formation of $I_2$ encompasses more substantial structural changes than that of $I_1$. The third SADS ($SADS_3$) exhibits a prominent negative peak in the small-angle region and has significantly larger intensity than $SADS_1$ and $SADS_2$, implying that the formation of $I_3$ involves relatively larger structural changes than the formations of $I_1$ and $I_2$, and major structural changes occur in the $I_2 \rightarrow I_3$ transition. The time-dependent population of SADSs shows that the population of $I_2$ increases as that of $I_1$ decreases with a time constant of 130 µs, followed by a decrease of $I_2$ as the population of $I_3$ simultaneously increases with a time constant of 3.4 ms (Fig. 2c).

Considering that $I_3$ was the intermediate formed last by the light activation in the sequential kinetic model, $I_3$ was predicted to correspond to the photoproduct (P). To validate this, we conducted static X-ray scattering (SAXS) experiments on both the ground (G) and light states of $As$LOV2 (See Methods and Supplementary Table 2 for details). A noticeable difference in the SAXS patterns emerged at the small-angle region before and after irradiation of the LED light (Supplementary Fig. 7). Notably, the static difference curve ($q\Delta S_{static}$, light state – G state) exhibits good agreement with $SADS_3$, indicating that $I_3$ observed in the TRXL measurement corresponds to P (Fig. 2b).

We collected the TRXL data on WT as well, which has a slower photocycle than I427V[48]. The small-angle region ($q < 0.2$ Å$^{-1}$) of the difference scattering curves shows a rise starting from 100 µs, with its magnitude more amplified after 17.8 ms. The wide-angle region ($0.2 < q < 0.6$ Å$^{-1}$) also shows a noticeable difference from 316 µs onwards (Supplementary Figs. 3 and 8). The difference scattering curves of WT are similar to those of I427V, except that the changes of WT occur on a slower timescale than I427V (Supplementary Fig. 3). We performed global kinetic analysis using SVD and KCA for the WT, following the same approach used for the I427V data, to investigate the kinetics and structural changes (Supplementary Fig. 8). From the SVD analysis, a sequential kinetic model, similar to that of I427V, with three transient species and two-time constants ($682 \pm 118$ µs and $10.6 \pm 2.89$ ms) was applied to WT. The kinetic model of WT exhibits relatively slower kinetics compared to that of I427V (Supplementary Figs. 8b–e). Various kinetic models consisting of three significant transient species and two-time constants were considered, and among them, the sequential kinetic model in which the three species are connected in series with two-time constants fits well with the experimental data.

Therefore, we can deduce that WT has the reaction pathway of $I_1 \rightarrow I_2 \rightarrow P$, as in I427V (Supplementary Fig. 8). Furthermore, the features and scattering amplitudes of the WT's SADSs obtained through KCA analysis are identical to those of I427V's SADSs within the experimental signal-to-noise ratio (SNR), indicating that the nature of structural changes involved in the light-induced transition of both constructs is the same (Supplementary Fig. 8b). In addition, there is no significant difference in the circular dichroism (CD) spectra between I427V and WT in the G state (Supplementary Fig. 2c), which indicates I427V mutation does not induce noticeable perturbation in the secondary structure of $As$LOV2 in the G state. The results from global kinetic analysis and CD data demonstrate that although the I427V mutation accelerates the photocycle of $As$LOV2[48], the mutation does not significantly affect the structural changes associated with the photocycle of $As$LOV2. These findings suggest that the light-induced transitions of both WT and I427V involve nearly identical structural changes. For this reason, we conducted structural analysis only on the I427V data, which has a relatively higher SNR compared to that of WT, and all the results of the TRXL-based structural analysis discussed in this study were derived from the I427V data.

## MD simulation-aided structural analysis of TRXL data

Previous studies on $As$LOV2 have suggested that the light-induced transition of this protein can be mediated by structural changes in the Jα helix[11,34,51–55]. Considering this structural information, two frames were taken into account for $As$LOV2, classifying its structure based on the structural characteristics of its Jα helix[11,34,51–55]: (i) a structure with a folded Jα helix, and (ii) a structure with an unfolded Jα helix. To extract the structural information from the three SADSs ($SADS_1$, $SADS_2$, and $SADS_3$) obtained from the experimental data, the structural frames of $As$LOV2 were simulated to generate various candidate structures as follows. For case (ii), to verify whether there is a structural preference due to the unfolding of the Jα helix, we performed non-equilibrium MD (NEMD) simulations for all physically allowed directions ($\pm x$, $\pm y$, and $+z$) through which the Jα helix could unfold (Supplementary Fig. 9). Through this process, we generated candidate structures with the Jα helix unfolded in various axial directions (details in Methods). In both simulations (i) and (ii), the crystal structure of G (PDB ID: 2V1A)[34] was used as the initial protein conformation. The MD-sampled structures obtained from these simulations were used to calculate theoretical X-ray static scattering curves corresponding to individual protein conformations. The theoretical difference scattering signal was generated by subtracting the static scattering curve of G from each theoretical static scattering signal. The theoretical scattering curve of G was selected from various theoretical curves obtained through MD simulations, based on the lowest $\chi^2$ value compared to the G state's SAXS curve. Subsequently, by fitting to minimize the $\chi^2$ values between theoretical and experimental difference curves, we selected the 10 best structures that best matched $SADS_1$ (or $SADS_2$ or $SADS_3$) (Supplementary Fig. 10). For each light-induced state ($I_1$, $I_2$, or P), the optimal structure that yielded the lowest $\chi^2$ value for the experimental data ($SADS_1$, $SADS_2$ or $SADS_3$) among the 10 best structures was used to illustrate the representative structural characteristics of the protein in each state. (details in Methods and Supplementary Fig. 10).

The best-fitted curves for $SADS_1$ and $SADS_2$ showed excellent agreement with the corresponding SADSs (Fig. 3a, b). A comparison between the optimal structure of $I_1$ and the G structure showed local structural differences between the two states (Fig. 3c). Only minor structural differences primarily in the Iβ and Jα were observed in $I_1$, as further indicated by the 2D difference distance map calculated between $I_1$ and G (Fig. 3c, e). On the other hand, the structural analysis of $SADS_2$ revealed that the optimal structure of $I_2$, and the 2D difference distance map between $I_2$ and G represent a fully unfolded Jα helix (Fig. 3d, f). Moreover, the structural analysis suggests that the theoretical difference curves better describe the features of $SADS_2$ when

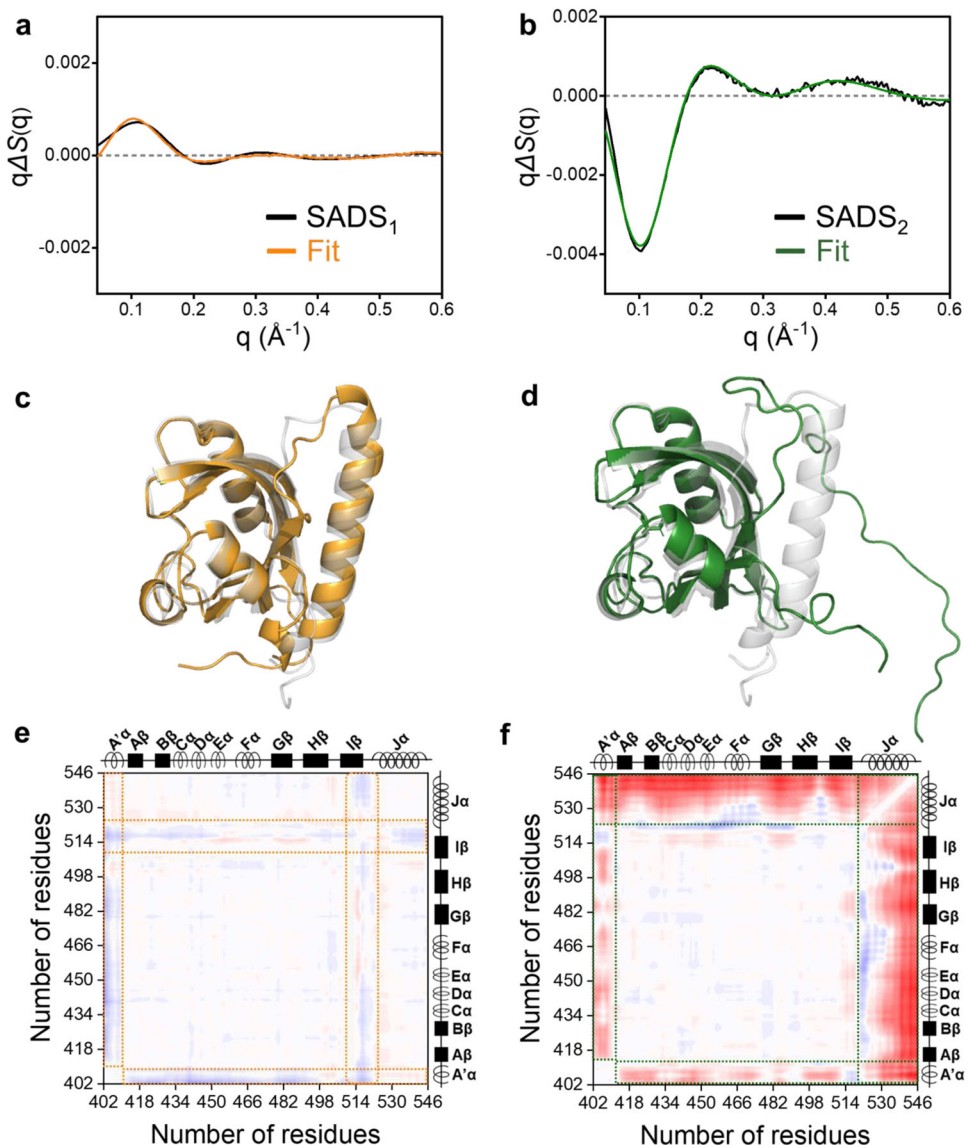

**Fig. 3 | Structural analysis of SADS$_1$ (corresponding to I$_1$) and SADS$_2$ (corresponding to I$_2$). a** Comparison of SADS$_1$ (black) and representative theoretical difference curves for I$_1$ (orange). **b** Comparison of SADS$_2$ (black) and representative theoretical difference curves for I$_2$ (green). **c** Optimal structure of I$_1$ (orange) that best describes SADS$_1$. **d** Optimal structures of I$_2$ (green) that best describe SADS$_2$. In (**c**) and (**d**), the G state structure (gray) is transparently displayed for comparison. **e,**  **f** Difference distance maps of (**e**) I$_1$ and (**f**) I$_2$ with respect to the G state. For each state, the distance map was plotted by calculating the difference of the corresponding Cα-Cα distances between the optimal structures and the G state structure. The regions where prominent distance changes are observed in the difference distance map are marked with square dashed boxes: orange in (**e**) and green in (**f**).

the Jα helix unfolds in the +y and +z directions, compared to other directions (±x and −y) (Supplementary Fig. 9). In addition, relatively minor structural differences around the A'α helix for these two states were also observed in the 2D difference distance map (Fig. 3f). To further confirm that the formation of I$_2$ is primarily mediated by the unfolding of the A'α and Jα helices, we conducted an additional control analysis. *As*LOV2 has a core structure comprising four α-helices, along with the A'α and Jα helices (Fig. 1b). Notably, the Fα helix is relatively similar in size to the Jα helix among these core helices. During the control analysis, we investigated the impact of the Fα helix unfolding on the formation I$_2$, with details described in Supplementary Note 1. The theoretical difference curves from the unfolded Fα helix structures failed to describe SADS$_2$, indicating that the Fα helix unfolding does not significantly contribute to the formation of I$_2$ (Supplementary Note 1 and Supplementary Fig. 11). This supports that the unfolding of the A'α and Jα helices is the primary structural change driving the formation of I$_2$.

Conversely, the structural analysis of SADS$_3$ could not yield satisfactory fits to SADS$_3$. To obtain a clue on the structure of P, we reconstructed molecular shapes based on static SAXS patterns of G and P (See Supplementary Methods and Supplementary Fig. 7b for details). A comparison of these two shapes indicates that a substantial global structural change is involved in the transition from G to P, as evidenced by the observation that the shape of G shows a radius of gyration (R$_g$) value of ~14.75 Å, whereas that of P shows an R$_g$ value of ~22.72 Å. Notably, the R$_g$ value of P is significantly larger than those of the optimal structures of I$_1$ and I$_2$ obtained from TRXL-based structural analysis (Supplementary Fig. 7c). Considering the prior structural studies of proteins sharing the LOV domain that have suggested the proteins may form a dimer in the light state in the light state[14,19,23–28], the significantly larger R$_g$ of P observed in our static SAXS and TRXL experiments compared to the three states observed (G, I$_1$, and I$_2$) indicates that *As*LOV2 can adopt a dimeric conformation when activated by light. Subsequently, we investigated whether the optimal

structures of G, $I_1$, $I_2$, and P could accurately describe the static scattering curve of the light state. The linear combination of the theoretical curves, calculated using the optimal structures, showed good agreement with the experimental static curve in the small-angle region (See Supplementary Note 2 and Supplementary Fig. 12 for details).

In addition to structural analysis, the SEC results also support dimerization conformation (See Supplementary Note 3, Fig. 4d, and Supplementary Fig. 13a for details). In the dark state, the SEC profile of WT only shows the monomer fraction, while in the light state, approximately 15% of the monomer fraction shifts to the elution volume corresponding to a dimer conformation. This shift confirms that the light-activated $As$LOV2 undergoes a structural transition from a monomer to a dimer, consistent with the findings from the scattering experiments.

For the structural analysis of $SADS_3$, we generated candidate structures with various dimeric conformations using MD simulations (Fig. 4a and Supplementary Fig. 14). To achieve this, we classified the previously reported dimer structure of LOV2 into three types of dimer conformations based on their dimeric interface[27,30,56]: (a) Structures where the A′α helices of two monomers (A′α–A′α) interact to form a coiled-coil dimerization, (b) structures where the β-scaffolds interact with each other (β-scaffold–β-scaffold), and (c) structures where A′α–A′α helices and Jα–Jα helices of two monomers interact to form coiled-coil dimerization. For each type of dimer conformation, we considered the following two monomer-excitation configurations: (i) homo L/L dimer (light state monomer + light state monomer) (a, b, and c frames) which consists of two light-activated monomers, and (ii) hetero L/G dimer (light state monomer + G state monomer) (a′, b′, and c′ frames) which consists of one light-activated monomer and one non-activated G state monomer. Consequently, we generated candidate structures by considering a total of six frame conformations through MD simulations (details in Methods).

The MD-sampled structures with the dimer conformations were used to calculate theoretical static scattering curves, which were then utilized in the structural analysis of P based on the approach used for $I_1$ and $I_2$. The residuals between each of the six conformations and $SADS_3$ show that the three homo L/L dimer configurations appear to better describe $SADS_3$ than those with hetero L/G dimer configurations (Fig. 4a and Supplementary Fig. 14). The optimal model with the lowest residual, obtained for the b-frame structure, well matches the molecular shape of the P obtained from static SAXS, supporting the validity of the structural analysis against $SADS_3$ (Fig. 4b). In the optimal structure of P, the Jα and A′α helices of each monomer subunit are unfolded, exposing the β-scaffolds, and the dimeric interface between the two subunits is formed on the exposed β-scaffolds (Fig. 4c).

To obtain detailed spatiotemporal aspects of the light-induced dimerization, we inspected the locations and trajectories of the intermediates with respect to the ground state across various parameter landscapes (Fig. 5). First, we examined the $R_g$ value as a function of the maximum particle dimension ($D_{max}$) based on the structural pool and optimal structures of each state (Fig. 5a). The trajectory in this parameter landscape (termed as L1) shows the gradual increases in the $R_g$ and $D_{max}$ values of the protein during the transition from G to P via $I_1$ and $I_2$ (Fig. 5a). Specifically, subtle changes are observed in the G → $I_1$ transition, while substantial changes occur in the $I_1$ → $I_2$ → P transitions, suggesting that the formation of $I_1$ involves minor structural changes, whereas the formations of $I_2$ and P are associated with more extensive structural changes. Furthermore, we considered the following three parameter landscapes (L2, L3, and L4): (L2) the distance between Ala523 and Ala543 (D1) as a function of distance between Phe403 and Glu412 (D2) (Fig. 5b), (L3) the solvent accessible surface area (SASA) among the β-scaffold, A′α and Jα helices (area 1 in Fig. 5d) as a function of root mean square deviation (RMSD) of the β-scaffold (Fig. 5c), and (L4) the SASA of area 1 as a function of the SASA among the β-scaffold and the remaining structural components of the

protein, excluding the A′α, Jα helices, and β-scaffold (area 2) (Fig. 5d). Specifically, for P, from each dimer, two locations were extracted from two monomer structures constituting the dimeric conformation of P (monomer A or monomer B; as depicted in Fig. 6), to facilitate the comparison of the landscape distributions of P to those of G, $I_1$, and $I_2$. Phe403 and Glu412 are located at the N-terminal and C-terminal of the A′α helix, respectively, while Ala523 and Ala543 are located at the N-terminal and C-terminal of the Jα helix, respectively. Therefore, in L2, the changes in D1 and D2 reflect the structural alterations in the A′α and Jα helices, respectively. L2 shows that the G → $I_1$ transition is characterized by an increase in D1, while the $I_1$ → $I_2$ transition demonstrates a decrease in D1 and an increase in D2 (Fig. 5b). This indicates that the formation of $I_1$ involves the structural extension of the A′α helix and that of $I_2$ includes both rearrangement of the A′α helix and unfolding of Jα helix. The optimal structure of $I_2$ shows that the rearrangement of the A′α helix corresponds to the unfolding of the helix. Specifically, the optimal structure shows that the unfolded A′α helix wound itself, and this is observed in the decrease of D1 as well. Additionally, the $I_2$ → P transition shows a slight increase in D1 and a broadening of the D2 distribution, suggesting that this transition involves rearrangements in both helices (Fig. 5a). In L3, during the G → $I_1$ transition, a substantial increase in the RMSD of the β-scaffold is observed, accompanied by a slight increase in SASA of area 1, implying that the transition is predominantly characterized by a structural change within the β-scaffold (Fig. 5c). For the subsequent $I_1$ → $I_2$ → P transitions, L3 demonstrates a significant rise in SASA of area 1 but not in the RMSD of the scaffold. In contrast to the notable changes in area 1 observed in L3, only a subtle change of SASA of area 2 is observed in L4 (Fig. 5d). These features indicate that the transitions involve the exposure of the β-scaffold to the external environment in the direction of the two helices (area 1).

## Discussion

The kinetic and structural analysis of the TRXL data show that the photoreaction of the $As$LOV2 domain has the kinetic model shown in Fig. 6, where it transitions from the G state to intermediate states $I_1$ and $I_2$, eventually leading to the formation of the photoproduct P. In this kinetic model, $I_1$, the first intermediate corresponding to $SADS_1$, is generated from the G state upon light activation. After that, $I_1$ is transitioned to $I_2$, the second intermediate corresponding to $SADS_2$, with a time constant of 682 μs for WT and 130 μs for I427V, and $I_2$ is finally transformed into P with a time constant of 10.6 ms for WT and 3.4 ms for I427V (Fig. 6). We compared the time constants observed in our kinetic analysis with those determined in spectroscopic studies (Tables 1 and 2). Time-resolved spectroscopic studies reported that in the photoresponse of $As$LOV2, FMN undergoes singlet-to-triplet conversion in the sub-nanosecond time regime, followed by the formation of cysteinyl–flavin (Cys–FMN) photoadduct of the protein in the time range of several μs to tens of μs[10,52–55,57–62] (Table 1). These primary events occur in a time range shorter than the temporal resolution of our study, which indicates that the transition from G to $I_1$ should involve the formation of the photoadduct. Regarding this, the absence of the SEC profile for a dimer conformation in the light state of the C450A mutant, in which the formation of the Cys–FMN photoadduct[10,63] is inhibited, confirms that the formation of the photoadduct is necessary for the light-induced dimerization (See Supplementary Note 3, for details). For the G → $I_1$ transition, we observe the increases in both D1 and the RMSD of the β-scaffold (Fig. 5b, c). Furthermore, the 2D distance difference map between the optimal structures of $I_1$ and G state, shown in Fig. 3e, also reveals that $I_1$ has two prominent structural differences in the Iβ and A′α helix compared to the G state. Considering the formation of the photoadduct, occurring within a time scale shorter than that of our TRXL measurement[10,52–55,57–62], these results suggest that the formation of the photoadduct mediates the subsequent structural changes of the

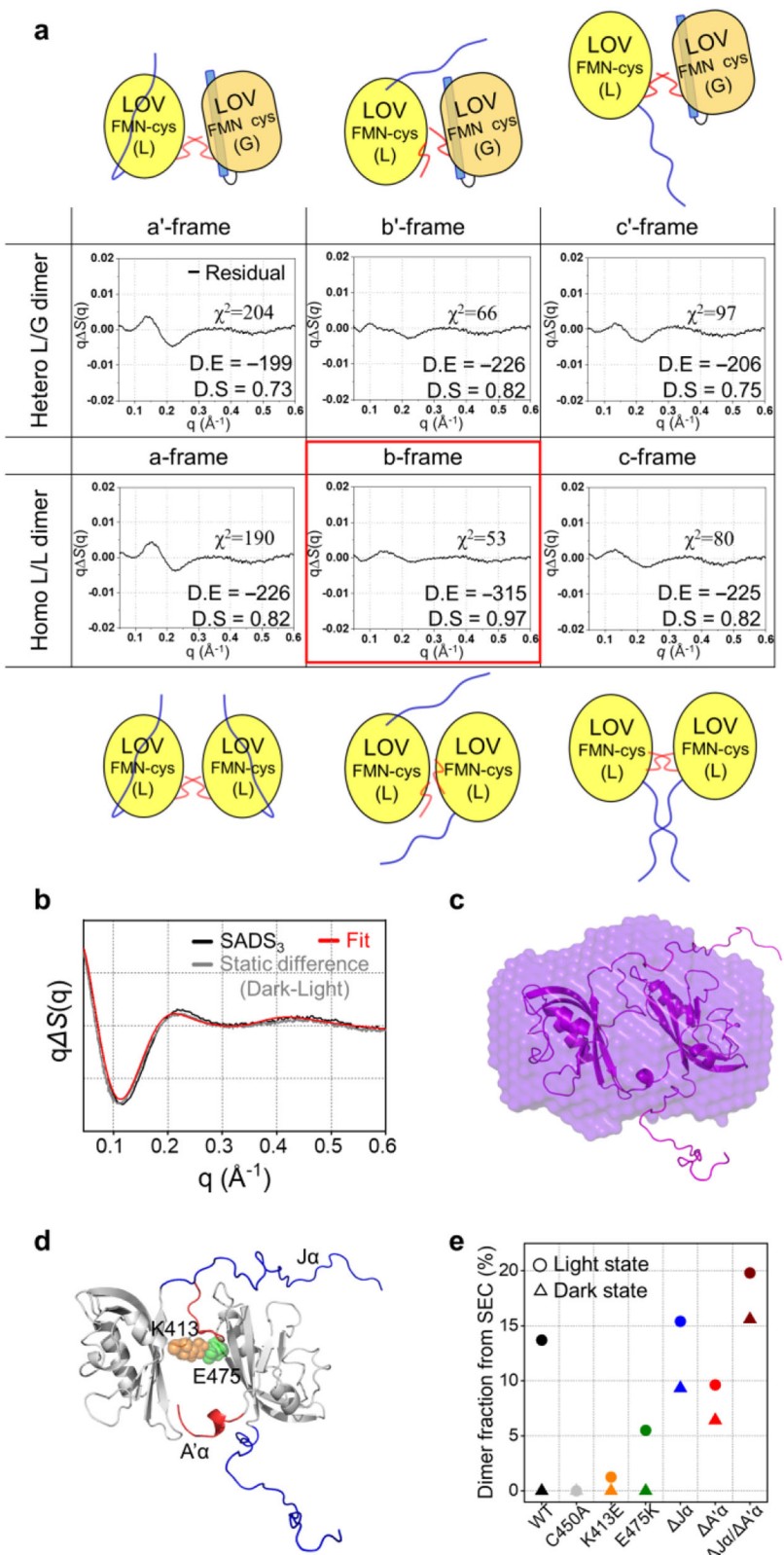

β-scaffold and A'α helix during the transition. Notably, the optimal structure of $I_1$ and the increase of D1 indicate that the formation of $I_1$ involves the structural extension of A'α helix[64] (Fig. 3c, 5b). Meanwhile, a previous structural study reported that the light-activated LOV domain from *Bacillus subtilis* (*Bs*YtvA) forms $I_1$ with a time constant of ~ 2 μs[47], which is similar to that observed for $I_1$ formation of *As*LOV2 in our study. Considering that *Bs*YtvA is a type of LOV

protein without the Jα helix, this similarity suggests that the Jα helix is not significantly involved in the formation of $I_1$ during the light-induced transition of LOV proteins. This is consistent with our structural analysis, which shows that structural changes in the A'α helix and β scaffold are involved in the formation of $I_1$.

The transition from $I_1$ to $I_2$ occurs with a time constant of 682 μs for WT, consistent with previous spectroscopy results that suggested a

**Fig. 4 | Structural analysis of SADS₃. a** Schematic diagram showing six representative *As*LOV2 dimer conformation candidates and the corresponding residuals between SADS₃ and the best-fit scattering curve determined from the structural analysis. In the upper panel, the results of structural analysis for three hetero L/G dimers are shown, and in the lower panel, those for three homo L/L dimers are shown. In the schematic of the frames for the homo- and hetero- dimers, the light state LOV2 core domain (L) is depicted in bright yellow, and the non-activated LOV2 core domain (G) is depicted in dark yellow. Jα helix is shown in blue, and A'α is shown in red. The best-fit curve with the smallest residual and χ² value is highlighted with the red box. The docking score (DS) and confidence score (CS) from molecular docking simulations, which assess the plausibility of dimer formation, are shown

for each conformation. **b** Comparison of SADS₃ (black), static difference scattering curves (qΔ$S_{static}$, light state – dark state, gray), and the best-fit curve of the optimal structure of P (red). **c** Comparison of low-resolution structures reconstructed from static SAXS and the optimal structure of P obtained from TRXL. **d** Representation of helices and residues associated with dimerization in representative protein conformations for P. Jα and A'α are shown in blue and red, and, K413 (yellow) and E475 (green) which form a salt bridge are represented as spheres. **e** Dimer fractions obtained from SEC for WT (black), C450A (light gray), K413E (orange), E475K (green), ΔJα (blue), and ΔA'α (red), and ΔJα/ΔA'α (brown). The SEC results for each construct in the dark state are represented as triangles, while those in the light state are represented as circles.

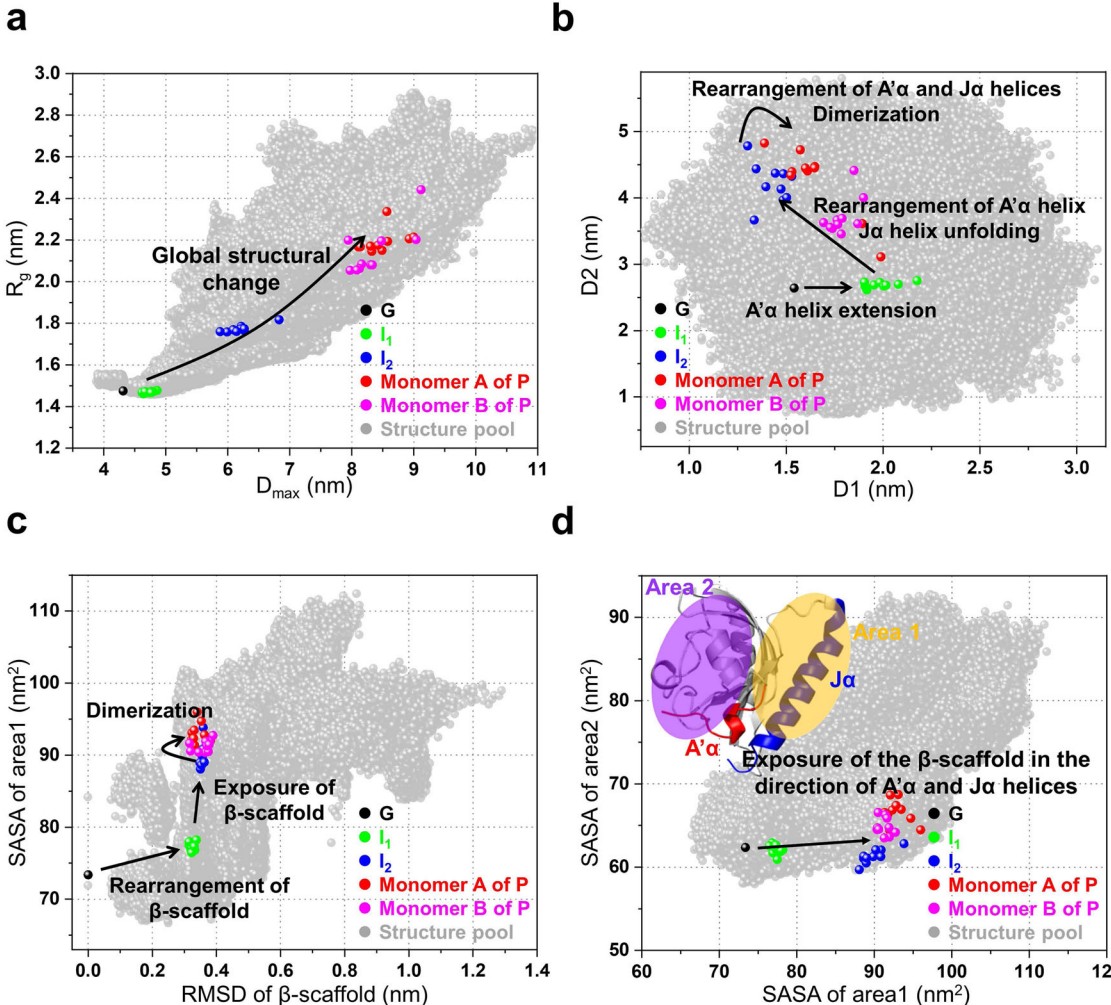

**Fig. 5 | Structural landscapes of the *As*LOV2 photocycle from the TRXL study. a** The landscape of $R_g$ as a function of $D_{max}$. **b** The landscape of distance between Ala523 and Ala543 (D1) as a function of distance between Phe403 and Glu412 (D2). Phe403 and Glu412 are located at the N- and C-terminals of the A'α helix, respectively, while Ala523 and Ala543 are located at the N- and C-terminals of the Jα helix, respectively. These distances track the light-induced structural changes of the A'α and Jα helices, respectively. **c** The landscape of solvent accessible surface area (SASA) among the A'α, Jα helices, and β-scaffold, (area 1 in the inset of Fig. 5d) as a function of root mean square deviation (RMSD) of the β-scaffold. The RMSD and SASA profiles unveil the β-scaffold's structural alterations and exposure of the β-scaffold to the external environment in the direction of the two helices, respectively. **d** The landscape of SASA of area 1 as a function of the SASA among the β-scaffold and the remaining structural components of *As*LOV2 excluding the A'α, Jα

helices, and β-scaffold (area 2 in the inset). In the structure (inset), A'α, Jα helices, area 1 and area 2 are denoted by red, blue, yellow and purple, respectively. The SASA profile of area 2 demonstrates changes in the exposure of the β-scaffold to the external environment in the direction of the remaining structural components. For each case of I₁ and I₂, the parameters were determined from the 10 best structures obtained in the structural analysis. For P, from each dimer, two parameters were calculated from two monomer structures constituting the dimeric conformation (monomer A or monomer B in Fig. 6) based on the 10 best structures of P. The values were calculated from the optimal structure of the G (black dots), and these values were compared with those of I₁, I₂, and P. The parameters for I₁, I₂, monomer A, and monomer B are denoted by green, blue, red, and magenta dots, respectively. The parameters calculated from the entire structural pool used in the structural analysis are represented in the landscapes (gray dots).

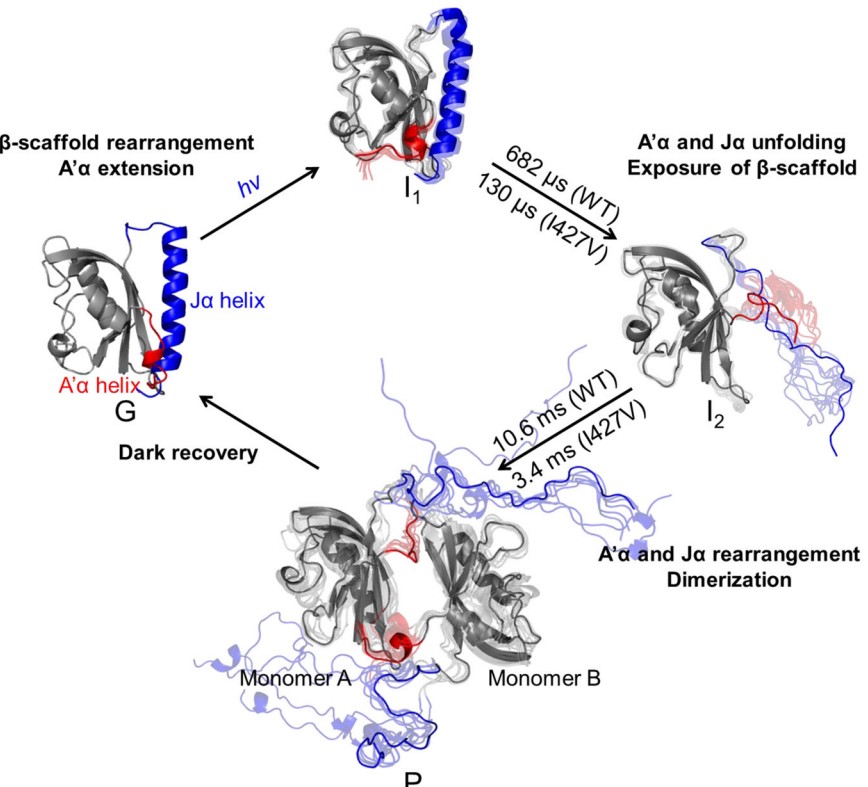

**Fig. 6 | Structural dynamics of the *As*LOV2 photocycle revealed by TRXL study.** The photocycle of *As*LOV2 includes the G state, three intermediates ($I_1$, $I_2$, and P), and related time constants (WT: 682 µs and 10.6 ms, and I427V: 130 µs and 3.4 ms), as determined from the kinetic analysis of the scattering data. For the TRXL measurements of WT and I427V samples, each sample (WT or I427V) was prepared at a concentration of 1.5 mM in 20 mM Tris pH 7.0, 200 mM NaCl buffer. The optimal structure of G was determined from the structural analysis on the static solution scattering data, while the optimal structures of three intermediates were extracted through the structural analysis using the TRXL data. The optimal structures ($I_1$ and $I_2$) indicate that the structural changes within the A'α and Jα helices allow the exposure of the β-scaffold to the external environment. Subsequently, *As*LOV2 undergoes dimerization (P), utilizing the dimeric interface formed between their β-scaffolds. For each state, the optimal structure is depicted alongside the 9 best structures, with the optimal structure highlighted and the remaining structures shown semi-transparently. The A'α and Jα helices are marked in red and blue, respectively, and the remaining structures are marked in gray.

transition on the timescale of tens to hundreds of µs[52–55] (Table 1). The $I_1 \rightarrow I_2$ transition accompanies the changes in D1 and D2, indicating that it involves the structural rearrangement of A'α and Jα helices. The optimal structure of $I_2$, where the A'α and Jα helices are unfolded, shows that the transition accompanies the unfolding of the helices (Figs. 3b, d, f, 5b, c). Specifically, our structural analysis shows that structures with unfolding Jα helix along the +y and +z directions better describe SADS$_2$ compared to those with unfolding along the other two directions (±x and −y), suggesting the existence of preferred directions of structural changes during the Jα helix unfolding in the transition (Supplementary Fig. 9). The unfolding of the Jα helix along the +y and +z directions increases the distance between the β-scaffold and the helix, leading to a greater separation between the Jα helix and the β-scaffold in the optimal structure of $I_2$ compared to G and $I_1$ (Supplementary Fig. 9). Considering that the SASA of the β-scaffold increases within area 1, not area 2, during the formation of $I_2$ (Fig. 5c, d), these structural characteristics suggest that the β-scaffold in $I_2$, which is shielded from the external environment by its interaction with the A'α and Jα helices in G and $I_1$, is no longer protected. This is consistent with previous studies that suggested disruption of the interaction between the β-scaffold and the Jα helix upon unfolding[11,51,65,66]. Furthermore, for the interaction between the helices and β-scaffold, the SEC profiles from constructs with the Jα helix deleted (ΔJα), the A'α helix deleted (ΔA'α), and both Jα and A'α helices deleted (ΔJα/ΔA'α) show the elution volume for the dimer conformation even in the dark state (Supplementary Figs. 13d–f), consistent with previous studies from several LOV domains from other species with truncated A'α or Jα

helices[18,64,67]. The docking simulations of the constructs demonstrate that the dimerization is more favorable in ΔJα/ΔA'α compared to those where only a single helix (ΔJα or ΔA'α) was removed (See Supplementary Note 3, Supplementary Methods, and Supplementary Table 1 for details). These results demonstrate that the intra-monomer interactions between the helices and β-scaffold hinder *As*LOV2 dimerization, supporting that the disruption of this interaction, as identified in our structural analysis, must precede *As*LOV2 dimerization.

Subsequently, $I_2$ converts into P with a time constant of 10.6 ms for WT. The structural analysis of P shows that P has a dimer conformation, indicating that the $I_2 \rightarrow P$ transition accompanies the association of two $I_2$ monomers to form a dimer. In this regard, the low-resolution structure of P obtained from our static SAXS data shows good agreement with the structure of P obtained from the structural analysis of SADS$_3$ (Fig. 4c), further confirming that *As*LOV2 dimerization is associated with the $I_2 \rightarrow P$ transition. In particular, in each monomer that constitutes the optimal structure, the Jα and A'α helices are unfolded, and the β-scaffolds form a dimeric interface (Fig. 4). Regarding this, in our structural analysis, candidates with the homo L/L dimer conformations exhibit a better fit to SADS$_3$ when compared to those with hetero L/G dimer conformations. In the homo L/L dimer conformation, the unfolded A'α and Jα helices of both monomers in the light state allow the β-scaffolds to freely interact with the external environment (Fig. 4a). In the hetero L/G dimer conformation, the monomer in the light state permits β-scaffold interaction with the external environment as the A'α and Jα helices unfold, whereas the monomer in the G state maintains intramolecular interaction between

**Table 1 | Comparison of time constants of *As*LOV2 observed in spectroscopic studies and TRXL**

|  | TR-IR[a] | TR-CD[b] | TA[c] | TRXL (This work)[d] | |
|---|---|---|---|---|---|
| Sample | WT | WT | WT | WT | I427V |
| FMN singlet-to-triplet conversion | 2 ns[54,61] 2.3 ns[53,54] | 3.3 ns[55] | 2.1 ns[62] | | |
| Cys-FMN photoadduct formation | 10 μs[52,54,61] 23 μs[53] | 4 μs[55] | 1.9 μs[62] | <5.62 μs | <5.62 μs |
| Jα helix partial unfolding | | 90 μs[55] | | | |
| Jα helix fully unfolding | 240 μs[52] 313 μs[53] > 500 μs[54] | | | 682 μs | 130 μs |
| Dimerization | | | | 10.6 ms | 3.4 ms |

[a]Time-resolved infrared spectroscopy.
[b]Time-resolved optical rotatory dispersion.
[c]Transient absorption spectroscopy.
[d]This work was conducted using time-resolved X-ray liquidography (TRXL).

**Table 2 | Comparison of time constants associated with LOV dimerization in various photoreceptor proteins**

|  | Aureo–LOV | Phot–LOV | *EL*222 | *As*LOV2 (This work) | |
|---|---|---|---|---|---|
| Dimerization time constant | 160 ms[a] 5 s[b22] 18 s[c16] | 6 ms[d20] 17 ms[e19] 40 ms[f18] 50 ms[g20] 1.7 s[h20] 2.9 s[i20] | 13 ms[j14] 230 ms[k21] | 10.6 ms (WT) | 3.4 ms (I427V) |

[a]250 μM *Vf*Aureo1-LOV-bZIP WT.
[b]20 μM *Pt*Aureo1a-bzip-LOV.
[c]*Vf*Aureo1-LOV-bZIP mutant.
[d]200 μM *Cr*phot-LOV1-hinge.
[e]50 μM *At*phot2-LOV1.
[f]50 μM *At*phot1-LOV2.
[g]80 μM *Cr*phot-LOV1.
[h]200 μM *Cr*phot-LOV2-hinge.
[i]200 μM *Cr*phot-hinge-LOV2.
[j]530 μM *EL*-LOV.
[k]475 μM *EL*222.

the β-scaffold and the two helices, thereby shielding the β-scaffold from exposure to the external environment (Fig. 4a). These structural differences indicate that the formation of dimeric interaction between the β-scaffolds of two different *As*LOV2 monomers is more favorable in the homo L/L dimer conformation compared to the hetero L/G dimer conformation. Considering that candidates with the homo L/L dimer conformation provide a better representation of SADS₃ than those with the hetero L/G dimer conformation, these structural differences between homo- and heterodimers demonstrate the necessity of freely interacting β-scaffolds for the formation of transient dimers in the light-induced transition of *As*LOV2. This key feature illustrates the critical role of interactions between the β-scaffold and the two helices in forming the dimer structure of P[27]. For the dimer conformations, docking simulations also indicate that the formation of dimers is more favorable in the homo conformations than in the hetero conformations (Fig. 4a and Supplementary Table 1), aligning with the results of our structural analysis (See Supplementary Note 3, for details). These structural characteristics suggest that the formation of I₂ should involve the unfolding of the two helices, exposing the β-scaffold to the external environment, which can lead to *As*LOV2 dimerization through the dimeric interaction formed between the β-scaffolds of two different monomers.

Additionally, the I₂ → P transition displays the change in the D1 and D2 landscape (L2), implying that the unfolded A'α and Jα helices undergo further structural arrangements during the formation of P

(Fig. 5a). Regarding this, the increases in the SASA of the β-scaffold within area 1 during the transition indicate that these arrangements lead to a more pronounced exposure of the β-scaffold to the external environment (Fig. 5c, d). Considering that the dimeric interface of P is formed between the β-scaffolds, these features suggest that the further arrangements of the unfolded helices facilitate intermolecular interactions between the β-scaffolds, thereby allowing for the more efficient formation of the dimeric interface in P.

For the light-induced dimerization, the structural landscapes also suggest that the light-induced dimerization of *As*LOV2 involves a gradual increase in the structural flexibility of the protein. L2 and L3 reveal gradual increases in the fluctuations of D1, D2, and the RMSD of the β-scaffold, indicating increasing structural flexibility of the Jα and A'α helices, as well as the β-scaffold, as the reaction proceeds. Notably, the fluctuation in the RMSD of the β-scaffold was smaller than those of D1 and D2, suggesting that the flexibility of the helices increases significantly more than that of the β-scaffold. Regarding this, the comparison of the 10 best structures of P shows differences in the overall structure, with significant structural variations in the Jα and A'α helices (Supplementary Fig. 10f). Considering that the helices are located on the surface of *As*LOV2, it is suggested that the global structure of *As*LOV2 would be flexible in the P state due to the structural flexibility of the helices. This global structural flexibility is also observed by L1, which shows the gradual increases in the fluctuations of $R_g$ and $D_{max}$ (Fig. 5a).

Moreover, it was predicted that electrostatic interaction between K413 near A'α helix and E475 in the β-scaffold is also related to the formation of the dimeric interface from the theoretical approach based on PDBePISA[50], an effective tool for identifying interactions at protein interfaces (See Fig. 4d and Supplementary Methods for details). Regarding this interaction, in the light state, the SEC profiles of K413E and E475K mutants, constructed to disrupt the electrostatic interaction, show a decreased dimer fraction compared to that of the WT (Fig. 4e and Supplementary Fig. 13b, c), confirming that not only the interaction between the β-scaffolds but also such electrostatic interaction plays a role in the formation of the dimeric interface (See Supplementary Note 4 for details).

Some spectroscopy studies reported that LOV domains found in other photoreceptor proteins, such as phototropin LOV domain (phot–LOV), *Erythrobacter litoralis* 222 (*EL*222), and aureochrome LOV domains (Aureo–LOV), form dimers on a relatively broad timescale ranging from ms to tens of seconds upon activation[14,16–19,21,22] (Table 2). In particular, the dimerization of phot–LOV proteins, such as the *Arabidopsis thaliana* phot1-LOV2 (*At*phot1-LOV2) and *Chlamydomonas reinhardtii* phot-LOV1 (*Cr*phot-LOV1) have been observed on a timescale of tens of ms, which is similar to the dimerization constant of 10.6 ms observed in this TRXL study[18–20]. Considering that *As*LOV2 is a type of phot-LOV, this similarity suggests that the dimerization of the LOV protein moiety involved in the light-induced transition of the phototropin superfamily shares similar kinetic properties[18–20] (Table 2).

For LOV proteins without an effector domain (short LOV domains), previous structural studies have shown that LOV domains within these proteins exhibit various conformational changes, including dimerization and dissociation, during light-induced transitions[23,32]. These structural changes enable the LOV proteins to interact with their partner proteins or molecules and mediate signal transduction processes. In contrast, LOV domains in LOV proteins with an effector domain exhibit structural changes that are synchronized with the modulation of the domains[14,28]. For several LOV proteins with an effector domain, it has been observed that the β-scaffolds in their LOV domains form the primary dimerization interface, implying a shared structural characteristic among LOV domains concerning their dimeric interfaces[14,16,17,21,22,27,67,68]. Furthermore, previous studies on LOV domains of phototropin proteins, a type of LOV domains found in LOV proteins with an effector domain, have suggested that the light-activated LOV proteins exhibit structural changes in their β- scaffolds[18,19,45], which strengthen their dimeric interactions. These

structural changes have implied that the enhanced dimeric interactions, in turn, facilitate the activation of downstream serine/threonine kinase domains by promoting structural rearrangement of the downstream moieties in phototropin proteins[45,69,70]. Considering these findings and the fact that the effector domain of *As*Phot1 is connected to the C-terminus of its LOV2 domain (*As*LOV2), the structural mechanism proposed in our study (Fig. 6) suggests that the light-induced dimerization of *As*LOV2 influences downstream reactions by reorganizing the structural arrangement of the effector domains and enhancing the interaction between them in *As*Phot1 (Figs. 1a and 6). Our study provides a structural interpretation of *As*LOV2's light-induced dimerization, as summarized in Fig. 6, which offers a conformational template for understanding how LOV proteins with an effector domain utilize light to regulate the structural and functional properties of their effector domains. The structural mechanism of the dimerization for *As*LOV2 suggests that protein oligomerization can be mediated by oligomeric interfaces facilitated by structural changes in subunits.

In this study, we elucidated the structural changes associated with *As*LOV2's light-induced dimerization using TRXL-based structural investigations, considering both local and global structural aspects. Kinetic analysis of TRXL data revealed that *As*LOV2 undergoes the formation of a photoproduct through two different structural intermediates induced by light activation (Fig. 2). Furthermore, our MD simulation-aided structural analysis demonstrated that light-activated *As*LOV2 converts to the photoproduct of transient dimer by generating the dimeric interface by the exposure of the β-scaffold due to the unfolding of the A'α and Jα helices (Fig. 6). The results of this structural analysis are also supported by additional SEC experiments and docking simulations, which show that the dimeric interaction of the protein is strengthened in the helix-deleted constructs (ΔJα, ΔA'α, and ΔJα/ΔA'α). Particularly, inspired by the structural information on *As*LOV2's light-induced dimerization transition, we propose that structural changes in the protein can mediate the formation of the oligomeric interface, ultimately leading to protein association. These findings will provide not only structural information involved in the light-induced dimerization of proteins sharing the LOV domain but also mechanistic insights into how proteins regulate their oligomeric states through protein association. Although our experimental and theoretical approaches successfully captured the structural changes associated with the formation of the dimer conformation for *As*LOV2, additional studies with improved time resolution and SNR, such as those employing serial crystallography, are necessary to elucidate the local structural changes involved in the photoadduct formation between the chromophore and the core domain of the protein, as these changes are considered to be crucial structural components for the light-induced structural transitions. Furthermore, in the LOV-based optogenetic field, the unfolding phenomenon of Jα has been primarily utilized to develop it as an optogenetic tool[71,72]. Our observation suggests the possibility of regulating dimerization more tightly and expanding the scope of dimeric optogenetic tools.

## Methods

### Preparation of *Avena sativa* LOV2 (*As*LOV2) samples

The wild-type (WT) *Avena sativa* LOV2 (*As*LOV2) protein was prepared following a reported protocol[11]. The coding sequence of *Avena sativa* phototropin 1 LOV2 domain (residue 404-546) was cloned into a 2B-T vector using the LIC (ligation independent cloning) method. The Jα helix deletion construct (ΔJα, residue 404-520), A'α helix deletion construct (ΔA'α, residue 413-546), and A'α and Jα deletion construct (ΔJα/ΔA'α, residue 413-520) were also cloned in the same manner. The mutants of *As*LOV2 (I427V, K413E, E475K, and C450A) were generated using a mutagenesis kit (EZchange Site-directed Mutagenesis kit, Enzynomics, South Korea). The primer sequences used for constructing *As*LOV2 variants are provided in Supplementary Data 1. *As*LOV2 WT, mutants, and variants were over-expressed in *Escherichia coli*

BL21(DE3) induced with 0.4 mM IPTG and purified using Ni-affinity chromatography (Ni-Sepharose 6 Fast Flow, Cytiva, USA) and an anion exchange chromatography system (Q-Sepharose Fast Flow, Cytiva, USA). The purified proteins were dialyzed against 20 mM Tris pH 7.0, and 200 mM NaCl. The concentrations of *As*LOV2 samples were determined by a UV-Vis spectrometer (UV-2550, Shimadzu, Japan).

### Time-resolved X-ray liquidography experiment and data processing

Time-resolved X-ray liquidography (TRXL) experiment of *As*LOV2 wild type (WT) was performed at the BioCARS 14-ID-B beamline of Advanced Photon Source (APS) (Argonne, USA) following the well-established experimental protocol[15,35–47]. The X-ray probe contained 24 single bunches of polychromatic 12 keV X-rays (standard top-up mode of APS), with a focusing size of 35 μm × 35 μm. After the irradiation of the protein sample by a circularly polarized nanosecond (ns) laser pulse (pump) with a fluence of 1.0 mJ/mm² at 450 nm, the X-ray probe pulse was delivered to the protein sample contained in a capillary flow cell to observe the light-induced structural changes recorded in small-angle X-ray scattering (SAXS) and wide-angle X-ray scattering (WAXS) regions. The X-ray scattering patterns were collected using a two-dimensional detector (MX340-HS, Rayonix, USA) for 20 and 18 time delays for WT and the I427V mutant, respectively. For the measurements, the WT and I427V samples (1.5 mM concentration) dissolved in 20 mM Tris pH 7.0, and 200 mM NaCl buffer were used. In order to prevent irradiation damage to the protein samples from X-ray and laser exposures, the sample solution was passed through a capillary flow cell using a syringe pump (LEGATO® 111, KD Scientific, USA), and a fresh sample was provided for every exposure of the pump and probe pulses. The measurements covered a time range from 5.62 μs to 316 ms for WT and from 5.62 μs to 100 ms for I427V. The scattering pattern at a negative time delay (−5 μs) is also collected and used as a reference for calculating the differences in the scattering curves. The scattering patterns at positive time delays contain structural information contributed by the initial state, intermediates, and photoproduct, while the scattering pattern at the negative time delay contains structural information of the initial state. From the scattering curves, the difference scattering curves, $\Delta S(\mathbf{q}, t) = S(\mathbf{q}, t) - S(\mathbf{q}, -5\,\mu s)$, were calculated by subtracting the scattering curve at negative time delay (−5 μs) from the curves at positive time delays. To better describe scattering features in the WAXS region ($q > 0.2\,\text{Å}^{-1}$), $q\Delta S(\mathbf{q}, t)$ was generated by multiplying q with $\Delta S(\mathbf{q}, t)$. For each time delay, more than 100 scattering curves were averaged to achieve a high signal-to-noise ratio. During the measurements, the temperature of the samples was kept at 293 K using a cold nitrogen stream (Oxford Cryostream, UK). From the TRXL data, the contribution from the heating of the solvent due to energy transfer by excited protein molecules was removed using a well-established protocol[39] (See Supplementary Methods and Supplementary Fig. 15 for the details).

### Global kinetic analysis for TRXL data (SVD and KCA analysis)

To identify the kinetics of the structural transition in *As*LOV2, we performed singular value decomposition (SVD) analysis on the TRXL data (difference scattering curves)[37,39]. SVD analysis decomposes the data matrix, A, of the difference scattering curve into three matrices, U, S, and V, satisfying the following equation ($A = USV^T$). U is a matrix called left singular vectors (LSVs), which is composed of time-independent spectra for each **q**. S is a diagonal matrix that represents the weight of each singular vector. The right singular vectors (RSVs) referred to as V are composed of time-dependent profiles for each LSV. LSVs offer a basis for the space spanned by species-associated difference scattering curves (SADSs), and RSVs present information on the time-dependent concentration changes of intermediates. Therefore, SVD analysis provides the number of structurally distinguishable intermediates and the dynamics of each species,

regardless of the kinetic model. Based on the singular values and autocorrelation factors of the corresponding singular vectors from SVD analysis, we identified three significant singular vectors (Supplementary Fig. 4). To determine the number of time constants involved in the light-induced structural transition, we simultaneously fitted the first three RSVs using: (i) exponential sharing one common time constant, (ii) exponentials sharing two common time constants, and (iii) exponentials sharing three common time constants (Supplementary Fig. 5). For both WT and I427V, the analysis using (ii) effectively described the RSVs features, while the analysis using (i) was insufficient, and the analysis using (iii) resulted in overfitting, confirming the involvement of two common time constants in the light-induced structural transition of WT and I427V. For the WT, the time constants of 682 µs and 10.6 ms were obtained within the time range of 5.62 µs to 316 ms. For I427V, time constants of 130 µs and 3.4 ms were obtained within the time range of 5.62 µs to 100 ms (Supplementary Figs. 3 and 8).

For each protein construct (WT or I427V), we employed kinetics-constrained analysis (KCA) to extract three species-associated difference scattering curves (SADSs) corresponding to the three species from the experimental scattering curves (Fig. 2b and Supplementary Fig. 8). In the KCA, we extracted SADSs from the kinetic components, specifically the major RSVs and LSVs, obtained from the SVD analysis rather than from the raw experimental scattering curves by using well-established method[73]. During the KCA the experimental scattering curves were decomposed into three SADSs corresponding to the three species as follows.

$$\Delta S_{theory}\left(q_i, t_j\right) = \sum_{k=1}^{3}\left[C_k\left(t_j\right)\right]\Delta S_{C_k}(q_i) \tag{1}$$

where $\Delta S_{theory}(\mathbf{q}_i, t_j)$ is the theoretical difference scattering curve at given $\mathbf{q}$ and $t$ values, $\Delta S_{Ck}(\mathbf{q}_i)$ is the SADS containing structural information for the formation of the $k_{th}$ intermediate species at a given $\mathbf{q}$ value, and $C_k(t_j)$ is the instantaneous population of the $k_{th}$ intermediate at a given $t$ value. The population can be calculated using the time constants obtained from the SVD analysis. Subsequently, we minimized the discrepancy between the theoretical and experimental curves by applying the Nelder-Mead simplex algorithm.

## MD simulations-aided structural analysis of TRXL data

We conducted molecular dynamics (MD) simulation-aided structural analysis on the three SADSs to extract the three-dimensional protein conformations involved in the light-induced transition of *As*LOV2. To this purpose, we classified the structural information of monomer conformations of *As*LOV2 into the following two representative conformational frames based on the structural characteristics of their Jα helix[11,34,51-55]: (i) a monomer structure with a folded Jα helix, and (ii) a monomer structure with an unfolded Jα helix. Subsequently, we performed the MD simulations considering the two representative conformational frames ((i) and (ii)) to generate candidate structures for describing SADS₁, SADS₂ and SADS₃.

To generate the equilibrium structures for (i), we performed the MD simulation using a crystal structure with monomer conformation (PDB ID: 2V1A)[34] as the initial structure. The simulations were performed using GROMACS 2019.2[74] with the Charmm36 force field[75], in combination with the TIP3P water model. To describe the cysteinyl–flavin (Cys–FMN) photoadduct condition of the chromophores, the force field was modified according to a previous study[76]. The charge of the system was neutralized by adding sodium ions. The system was equilibrated under NVT condition for 100 ps with a velocity-rescale thermostat ($\tau_T = 2$ fs and $T = 300$ K) and was subsequently equilibrated under NPT condition for 100 ps with a velocity-rescale thermostat ($\tau_T = 2$ fs and $T = 300$ K) and a Parrinello-Rahman barostat ($\tau_T = 2$ fs and $T = 300$ K). After equilibration, 100 ns

production simulations were performed on the equilibrated structure. After the simulation, 10,000 candidate structures with (i) were sampled from the MD trajectory at 10-ps intervals. From the MD trajectories, the representative structure for (i) was selected by using the clustering method based on the GROMOS algorithm[77] and then used as the starting structure for the MD simulation of (ii).

To generate protein conformations with the geometry of (ii), we used the representative structure for (i) as an initial structure and conducted a non-equilibrium molecular dynamics (NEMD) simulation employing the pulling algorithm of GROMACS. In the pulling simulation, we defined two residues, Gln502 (Q1) and Leu546 (Q2), representing the N and C-terminus of the Jα helix. Afterward, we set the direction of the external pulling force to cause the Q2 position to move away from Q1, thereby inducing the unfolding of the Jα helix. Considering the structural flexibility of the unfolded Jα helix in (ii), we applied the force to pull the Jα helix along the directions of ±x, ±y, and z (Supplementary Fig. 9). In pulling simulations, the pulling force strength was applied as 1000 kJ/mol·nm² between two positions. All NEMD simulations were conducted using the pulling algorithm along the hypothetical reaction coordinates until the Jα helix completely unfolded. From each NEMD trajectory, 5000 candidate structures for (ii) were sampled at 10-ps intervals.

After the simulations, we aimed to find the optimal protein conformations for I₁, I₂, and P by comparing SADSs with the difference theoretical scattering curves from the MD-generated structures. We calculated the theoretical scattering curves of the candidate structures using the CRYSOL[78] with all default parameters. To account for the effect of the polychromatic X-ray beam, a well-established protocol was used to incorporate the energy spectrum of the X-ray pulse into the theoretical scattering curve[39]. Subsequently, the theoretical difference scattering curves were obtained by subtracting the theoretical static scattering curve corresponding to the ground (G) state from the theoretical static scattering curves corresponding to I₁, I₂, and P. After generating the candidate pools of theoretical difference curves, the theoretical difference scattering curves were compared with three SADSs to identify the optimal structures for I₁, I₂ and P. The optimal structures were assessed by calculating the reduced $\chi^2$ using the following equation, which quantified the discrepancy between the theoretical and experimental difference scattering curves.

$$\chi^2_{red} = \frac{1}{N-1}\sum_{i=1}^{N}\left[\frac{\mu \cdot \Delta S_{theory}(q_i) - \Delta S_{exp}(q_i)}{\sigma(q_i)}\right]^2 \tag{2}$$

$\Delta S_{theory}(\mathbf{q})$ represents the theoretical difference scattering curve, $\Delta S_{exp}(\mathbf{q})$ corresponds to the experimental TRXL data, $\mu$ is the scaling factor between $\Delta S_{theory}(\mathbf{q})$ and $\Delta S_{exp}(\mathbf{q})$, $N$ is the number of $\mathbf{q}$ points in the experiment, and $\sigma(\mathbf{q})$ represents the experimental standard deviation. $\mu$ comprises two scaling factors: (i) the relative ratio between the experimental and theoretical scattering curves, and (ii) the practical photoconversion yield of *As*LOV2 in the TRXL experiment. In principle, a $\chi^2_{red}$ value close to 1 indicates the most optimal solution within the experimental error. Based on the possible structural frames, for each state, the 10 best-fitted theoretical difference curves with the lowest $\chi^2_{red}$ values were selected as the best solutions. The 10 best protein structures were then selected from the MD-generated structures, which were used to generate the 10 best-fitted theoretical curves. These 10 best structures were subsequently used to generate the structural landscapes (Fig. 5), providing detailed structural information about the light-induced transition of *As*LOV2. Furthermore, for each light-induced state (I₁, I₂, or P), the optimal structure with the smallest $\chi^2_{red}$ value among the 10 best structures was used to illustrate the representative structural characteristics of the protein in each state (Figs. 3, 4, and Supplementary Fig. 10).

The optimal solutions for $SADS_1$ and $SADS_2$ demonstrated excellent agreement with the corresponding experimental data (Fig. 3), whereas the solution for $SADS_3$ did not. Regarding this, the low-resolution structure of P obtained from the ab initio shape reconstruction of the SAXS profile for the light state suggested that *As*LOV2 can potentially adopt a dimeric conformation when activated by light. Thus, we additionally performed MD simulations considering three representative conformational frames to generate candidate structures with various dimer conformations for describing $SADS_3$[27,30,56]: (a) Dimer structures where A'α helices of two monomers (A'α–A'α) interact to form a coiled-coil dimerization, (b) Dimer structures where β-scaffolds interact with each other (β-scaffold–β-scaffold), and (c) Dimer structures where A'α–A'α helices and Jα–Jα helices of two monomers interact to form coiled-coil dimerization. For each type of dimer conformation, we considered the following two monomer-excitation configurations: (L1) homo L/L dimer (light state monomer + light state monomer) (a, b, and c frames) which consists of two light-activated monomers, and (L2) hetero L/G dimer (light state monomer + G monomer) (a', b', and c' frames) which consists of one light-activated monomer and one non-activated G monomer. By considering the three dimer conformations and two monomer-excitation configurations, a total of six MD simulations were conducted to generate candidate structures with dimer conformations. For MD simulations of (L1), we used two optimal structures of $I_2$ as two monomers and aligned them according to the dimer geometries of (a), (b), or (c), thereby obtaining an initial structure for each dimer conformation (a, b or c) (Supplementary Fig. 14). For MD simulations of (L2), we generated an initial structure for each dimer conformation (a', b', or c') by using one optimal structure of $I_2$ as a monomer unit and one G structure as the other monomer unit and aligning them according to the dimer geometry of (a'), (b'), or (c') conformation (Supplementary Fig. 14). The initial structures were used for MD simulations, and these MD simulations were carried out in the same manner as the simulation applied for (i). During these MD simulations, production simulations were conducted over a period of 300 ns. 30,000 candidate structures with dimer conformations were sampled from each MD trajectory at 10-ps intervals. These structures were employed in the structural analysis for $SADS_3$ using the same approach as applied to the candidate structures with the monomer conformations, and the 10 best structures of P were obtained. The initial structures and optimal structures for each state from the MD simulations-aided structural analysis are provided in Supplementary Data 2.

## Reporting summary

Further information on research design is available in the Nature Portfolio Reporting Summary linked to this article.

## Data availability

The time-resolved difference scattering data analyzed in this study, corresponding to Fig. 2a, b, is provided in the Source Data file. The SAXS profiles for *As*LOV2 in dark and light states have been deposited in the SASBDB with the following entries: SASDV55 (dark state) and SASDV65 (light state). The sequence information for the primers used in the preparation of the protein samples is provided in Supplementary Data 1. Additionally, the initial and final structures for the light-induced structural changes of *As*LOV2, derived from MD simulations-aided structural analysis, and the SEC data for the *As*LOV2 constructs are provided as Supplementary Data 2 and Supplementary Data 3, respectively. The PDB code of the previously published structure used in this study is 2V1A. Source data are provided with this paper.

## Code availability

In this study, the MATLAB svd function was used for the SVD analysis, and the OriginLab exponential fitting function was used for exponential fitting. These functions can be utilized according to the manufacturer's manuals. The code for KCA analysis used in this study is available from the corresponding author upon request. For the MD-aided structural analysis, the open-source software packages (GROMACS 2019.2 and ATSAS), as well as the MATLAB fmincon function, were used. Additionally, docking simulations and the identification of potential interactions were conducted using the HDOCK and PDBePISA web servers, respectively.

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

## Acknowledgements

We thank Yonggwan Kim for his assistance with time-resolved X-ray liquidography experiments. We appreciate Min Byoung Seok and Kyeong Sik Jin, the 4 C SAXS II beamline staff at Pohang Accelerator Laboratory (PAL, Korea), for their assistance with SAXS data collection. This work was supported by the Institute for Basic Science (IBS-R033) granted to H.I. The use of the BioCARS Sector 14 was also supported by the National Institutes of Health, and the National Institute of General Medical Sciences grant R24GM111072. The time-resolved setup at Sector 14 was partially funded through collaboration with P. Anfinrud (NIH/NIDDK) through the Intramural Research Program of the NIDDK.

## Author contributions

C.K., S.R.Y., S.J.L., and S.O.K. contributed equally to this work. S.O.K. and H.I. conceived the idea. S.R.Y., S.O.K., and H.I. designed the research. C.K., S.R.Y., S.O.K., H.L., and S.Y. prepared the sample. C.K., S.R.Y., S.O.K., S.Y., H.L., T.N., and J.B. performed static SAXS experiments. C.K., S.R.Y., S.O.K., H.L., and I.K. performed TRXL experiments. C.K., S.R.Y., S.J.L., J.G.K., and H.I. analyzed the data. C.K., S.R.Y., S.J.L., S.O.K., H.L., J.C., J.G.K., T.W.K., and H.I. discussed the results. C.K., S.J.L., and H.I. performed the revision process. C.K., S.R.Y., S.J.L., and H.I. wrote the manuscript, and all authors approved the final version of the manuscript for submission.

## Competing interests

The authors declare no competing interests.
