## [Peer Review File · Nature Communications]

Structural dynamics of protein-protein association involved in the light-induced transition of *Avena sativa* LOV2 proteinREVIEWER COMMENTS

Reviewer #1 (Remarks to the Author):

In this paper the lhee team present excellent time resolved scattering experiments with very good results and interesting conclusions. I highly recommend publication of this interesting paper. I have a couple of questions and recommendations. The data is excellent quality, and the SVD analysis is done to very high standard. The MD simulations are in general appropriate, but I recommend some more critical discussion

Firstly the title is not very informative. It should refer to the LOV protein, and ideally the protein function and process studied.

The introduction refers to previous studies on a dimeric construct, please spell out these results in more detail to contrast them with the present results, if space allows.

Minor things and questions:

The abstract says “with spatiotemporal details that could not be accessible in previous studies” is a vague statement, please explain these previous studies here, and why

Page 12line 227 , please explain how many candidate structures were investigated

Page 18, the authors conclude that “formation of I2 should involve the unfolding of the two helices, exposing the β -scaffold to the external environment, which can lead to AsLOV2 dimerization through the dimeric interaction formed between the β -scaffolds of two different monomers”. Is it possible to discuss the strength of this conclusion please. It is difficult to follow some of the reasoning. Which changes are isolated from MD to support this. It would be helpful if the conclusions could more clearly summarise the strength of the evidence. This is important because conclusions arise from a combination of excellent quality experimental results, and MD simulations which are never guaranteed to be correct. Please try to phrase this as realistic as possible, I am suggesting to identify strengths and weakness for the correspondence between MD and experiment

SI page 13, figure 5. Is the data for the light state pure or is it extrapolated? If it's extrapolated then please fully explain the quantification. Has there been made an attempt to fit coordinates to this envelope

SI figs 5,6,7. As you know the low-q information is essential in order to properly fit the data and extract the correct contrast. I am surprised the SAXS beamline does not go to lower q range. Can you explain what limited the q range, and how zero q extrapolated amplitudes were estimated

Reviewer #2 (Remarks to the Author):

This manuscript reports on a light-activated protein-protein interaction mechanism in the LOV2 domain. The authors use time-resolved X-ray liquidography (TRXL) to monitor first – an unfolding event, which is then followed by a dimerization process. The proposed mechanism is substantiated by mutational studies and chromatography.

This manuscript presents very nice TRXL data – and the kinetic modeling and – in particular – the MD-based structural refinement approaches are impressive and contain novel features that can be transferred to other target systems. However, the biological aspect(s) should be elaborated upon to better highlight how the results reconcile with prior time-resolved studies of LOV proteins, and put the current results into perspective of LOV signaling in general (see more details below).

Major concerns:

* The Abstract and the Introduction are written as if this would be THE mechanism for PPIs – which is most likely not the case. I agree that it can be one out of many mechanisms. Therefore, the manuscript would benefit from a rewriting to present the observed results as THE mechanism for activation of this particular LOV protein. Such specification avoids overarching claims – but also strengthens the manuscript and makes it more to the point. And as a consequence, the Title should be revised.

* TRXL studies have been performed on LOV proteins e.g. Berntsson et al., Structure, 2017 (cited) & Henry et al., Biochemistry, 2020 (not cited). Since the timescales in part overlap with those in the current manuscript and the proteins are (very) closely related – this should be commented upon. E.g. similarities/differences in the protein origins & constructs, dS profiles, and structural interpretations.

* The underlying biology should be more elaborately described. For instance, how general is the mechanism identified in this work given that other LOV activation mechanisms have been presented, e.g. repositioning of LOV domains (Heintz et al., 2016, eLIFE) & monomerization (Conrad et al., 2013, Biochemistry). AND – what are the consequences for downstream signaling – which is not mentioned so much in the current version.

Minor concerns:

* Overall this is a well written manuscript, but a careful readthrough will likely correct a few phrasings/repetitions here and there throughout the manuscript.

* Sometimes “microsecond” is used and sometimes “ μ s” – same for “milliseconds” vs “ms” I believe.

* p. 4, l. 64 – I427V mutant is mentioned without citation – was this known before or somehow from this study? I guess that a citation is missing.

- * The Results start with the mutant data – which we learn later on (end of Results section) is because of better signal-to-noise ratio. This was confusing to me. Can WT and mutant be presented side-by-side? Then it even be more apparent that they are similar and still be logical.
- * Obvious to some but perhaps it should be mentioned somewhere that the P population sums to 0.5 due to dimerization?
- * Treatment of the TRXL data (SVD & kinetic modeling) is excellent and the TRXL ability to scrutinize between sequential and parallel models is impressive.
- * MD-based unfolding of J α helix to model SADS2 is interesting. However, it would be even more convincing if the authors could show that unfolding a helix somewhat corresponding in size (e.g. F α) would result in worse fits to the data. I.e. a repeat of Suppl. Fig. 8 but for helix F α .
- * 100 ns of MD simulations is very short on the biological time scale of dimerization. Therefore, while one E/E dimer with b-frame reproduced the data best, I wonder if any of the remaining models could have achieved an equally good fit given enough simulation time? Perhaps this can be answered by plotting RMSD over time. Seeing that the E/E dimer with b-frame has low RMSD throughout will indicate stability – and perhaps the other systems drift apart. In any case, RMSD vs time plots would be a nice addition.
- * Is Fig. 4d referred to somewhere?
- * p. 23 – how were the heating signals removed from the TRXL?
- * p. 25 – Gromacs citation missing.
- * p. 26 – which clustering method?
- * p. 26 – were all CRY SOL parameters default?
- * References – too many general references on PPI's 1-8 – maybe one or two is enough? Also, see major concern above about the focus of the manuscript.
- * Suppl. Fig. 1 – why not use the same concentrations in the SEC as in the SAXS experiments?

Reviewer #3 (Remarks to the Author):

The manuscript provides a detailed insight into the light-induced structural changes of a phototropin LOV domain, which is active in plant photoreceptors and in optogenetic tools. A series

of time-resolved SAXS and WAXS recordings are analyzed by integrating theoretical scattering curves from model-based molecular dynamics simulations. The analysis covers the time region of microseconds and milliseconds after which the photochemistry is concluded and the protein responses take place. The LOV domain is one of the most intensively investigated photoreceptor domains because of the small size, easy production and large conformational changes. Accordingly, many studies by time-resolved transient grating and infrared spectroscopy are available and summarized in Tables 1 and 2. The key advance of this study is the spatial resolution achieved by the combination of scattering experiments and MD simulations, in which different model assumptions for the protein were iteratively tested for an agreement with the experiment. The experiments were well conducted and analyzed and represent an impressive methodological achievement. However, novel insight into LOV domain structure and dimerization is limited compared to previous knowledge. The same method has been applied to LOV domains previously (refs 47 to 49) albeit without the present focus on light-induced dimerization. In conclusion, the study nicely summarizes and confirms previous insight into the mechanism of phototropin-LOV dimerization by other techniques but the specific additional insight is unclear.

1. Figure 3 represents a central figure of this manuscript and points to a single-residue specific spatial resolution. How realistic is such a resolution from analyzing SAXS/WAXS recordings? In other words: How large is the resolvable change in distance by illumination and how large is the respective error?
2. Key additional insight is obtained by studying truncated constructs of LOV without extending helices by SEC. The SEC peaks in Supp Figure 10 demonstrate some dimerization of LOV under illumination. The ratio of dimer to monomer is very low (15%) and at odds with time-resolved SAXS. Other studies on LOV have demonstrated a nearly 100% shift from monomer to dimer or to a dynamic equilibrium in the SEC profile (see for example Zoltowski and Crane 2008 *Biochemistry* 47 7012). At which wavelength was the detection of the SEC profiles performed? The scattering profiles seem to provide more solid evidence for the dimerization than the SEC. Accordingly it would be helpful to study truncated constructs by SAXS instead of SEC.
3. Figure 6. The depicted structural model of the final P state is puzzling because it infers a strongly extended and flexible structure of the helices which is not in agreement with the constraints from SAXS (Fig. 4c). D_{max} values should limit the extension of the dimer by defining the maximal average distance of scatter centers in the ensemble. The real structure should be an envelope of snapshots showing the flexibility of the helices in solution. Moreover, the chosen representative structures show a residual secondary structure in both helices. Is this residual helical structure only a snapshot in an otherwise unfolded scenario or a representative picture of the P state?
4. Table 1 shows a broad overview of previous time-resolved experiments on unfolding of helices in LOV domains. Results from the transient grating technique by Terazima and coworkers are not included although helix unfolding has been addressed by this scattering technique with precise time constants for different phototropin and aureochrome LOV domains (Refs 20-26). What is the reason for not including these results in Table 1?

5. Page 17. The finding that A α and J α helices must both unfold for dimerization is not new and should be supported by references to previous studies (Zayner et al. 2012 J Mol Biol 419 61; Takeda et al. 2013 J Phys Chem B 117 15606; Herman et al. Biochemistry 2015 54 1484), which used exactly the same approach of truncating the extending helices.

Minor comments:

- Abstract. Please use *Avena sativa* instead of *Avena Sativa*.
- Page 18. The denomination of light state as 'excited state' is misleading, because the real excited (triplet) state decays within a few microseconds prior to the experimental time window.
- Page 20. If a LOV protein is mentioned in the discussion, it must be stated whether this refers to phot1 or phot 2, LOV1 or LOV2.

Responses to the comments from Reviewer #1

In this paper the Ihee team present excellent time resolved scattering experiments with very good results and interesting conclusions. I highly recommend publication of this interesting paper. I have a couple of questions and recommendations. The data is excellent quality, and the SVD analysis is done to very high standard. The MD simulations are in general appropriate, but I recommend some more critical discussion.

→ We appreciate the positive evaluation of the reviewer.

Firstly the title is not very informative. It should refer to the LOV protein, and ideally the protein function and process studied.

→ We changed the title from “Structural dynamics of protein-protein association unveiled by time-resolved X-ray liquidography” to “Structural dynamics of protein-protein association involved in the light-induced transition of *Avena sativa* LOV2 protein”.

The introduction refers to previous studies on a dimeric construct, please spell out these results in more detail to contrast them with the present results, if space allows.

→ Regarding the reviewer's comment, we revised the introduction as follows. The parts with major changes are indicated in bold.

“Photoreceptors detect light signals, a vital environmental stimulus for living organisms, and convert them into biological signals. This enables the organisms to adapt to and respond to changes in their light environment. The LOV superfamily acts as photosensory units in various enzymes and signal transduction proteins with flavin mononucleotide (FMN) as their chromophores. The core structure of LOV domain consists of five central β -sheets (A β , B β , G β , H β , and I β) and four α -helices (C α , D α , E α , and F α), which are highly conserved throughout blue-light receptors (Fig. 1). Upon irradiation, LOV domains utilize FMN to sense blue light, converting such stimuli into structural signals through conformational changes in the protein moiety.

Various studies have been conducted on the photoresponse of these LOV domains. These studies have proposed that, upon activation by blue light, a photoadduct is formed through a covalent bond between FMN and cysteine located within the E α of the domains, followed by subsequent structural changes in the protein moiety of LOV domains. **Especially, photoreceptors that possess LOV domains as light-sensing regions undergo dimerization during their light-induced signal transduction. These structural changes play a crucial role in modulating downstream biological processes.**

Although spectroscopy studies have suggested that various LOV domains may form transient dimers, the dimerization observed in these studies was dependent on concentration, even without light exposure. Moreover, while ultrafast studies effectively probe the electronic structure of the chromophore, they do not directly detect global structural changes of the proteins involved in photo-induced dimerization. Notably, as summarized in Supplementary Figure 1, several structural studies on the proteins have reported the dimer conformations of LOV domains, suggesting that these proteins can adopt dimer conformations with various dimeric interactions. However, they do not offer

detailed structural insights into the mechanisms of the photo-induced dimerization. These limitations have made it challenging to obtain specific structural information about dimerization, leaving the structural characteristics of LOV domain dimers and their dimerization processes elusive.

In this work, to overcome these challenges, we apply time-resolved X-ray liquidography (TRXL) to capture the photoinduced structural dynamics in *AsLOV2*, which does not show concentration dependence on dimerization in the dark state (Fig. 1 and Supplementary Figs. 2a and 2b). *AsLOV2* (residue 413-520) belongs to the LOV2 domain of *Avena sativa* phototropin 1 (*AsPhot1*), sharing the same structural framework as canonical LOV domains. We used a construct that incorporates the A' α helix (residue 404-412) connecting to the LOV1 domain on one side and the J α helix (residue 521-546), which links to the serine/threonine kinase (STK) domain on the other side of *AsLOV2*. Due to the presence of the A' α and J α helices, this construct maintains a monomeric state in the dark, making it a suitable model system for observing transient dimerization upon irradiation. In this work, we refer to this construct including the A' α and J α helices (residue 404-546) as *AsLOV2*. **TRXL, also known as time-resolved X-ray solution scattering (TRXSS), is one of the sensitive experimental methods for detecting global conformational changes in solution induced by a trigger such as optical excitation and temperature jump. When TRXL is applied to macromolecules, it is also referred to as TR small-angle/wide-angle X-ray scattering (TR-SAXS/TR-WAXS).** Kinetic analysis of the TRXL data from both wild type (WT) and I427V mutant, the latter of which exhibits a relatively faster dark recovery compared to WT, identified three distinct intermediates connected by two time constants on the microseconds and milliseconds timescales in the light-induced structural transition of LOV2. Additionally, structural analysis aided by molecular dynamics (MD) simulations revealed that LOV2 undergoes extensive conformational changes related to the changes in oligomeric state (dimerization) through two monomeric intermediate states. Regarding the dimerization, we demonstrated that the formation of the dimeric interface for *AsLOV2* is associated with PPIs driven by the inter-monomeric interaction between the β -scaffolds, as evidenced by size exclusion chromatography (SEC) experiments on three *AsLOV2* constructs: a construct with the J α helix deleted (Δ J α), a construct with the A' α helix deleted (Δ A' α), and a construct with both J α and A' α helices deleted (Δ J α / Δ A' α). Moreover, we observed that the SEC profiles of K413E and E475K mutants, designed to disrupt the electrostatic interaction between *AsLOV2* monomers, exhibit a decreased dimer fraction compared to that of the WT. **These observations support that the PPIs, including not only the interaction between the β -scaffolds but also a local electrostatic interaction, contribute to the formation of the dimeric interface. The findings of our study provide direct insights into the structural mechanism of photo-induced dimerization in *AsLOV2*, as well as a structural template for understanding the process of protein-protein association driven by PPIs."**

We added **Supplementary Figure 1** to present diverse LOV dimer structures.

Supplementary Figure 1. Dimeric structures of LOV domains reported in the literature: *Arabidopsis thaliana* Phot1-LOV2 (*Atphot1-LOV2*, PDB ID: 4HHD), *Bacillus subtilis* LOV (*BsYtvA*, 4GCZ), *Neurospora crassa* VVD (*NcVVD*, 3RH8), *Phaeodactylum tricornutum* Aureochrome 1a-LOV (*PtAureo1a-LOV*, 6T74, 5A8B, 5DKL), *Rhodobacter sphaeroides* LOV (*RsLOV*, 4HJ3), and *Erythrobacter litoralis* 222 (*EL222*, 3P7N). The LOV domain structures are shown with the PAS core domain colored yellow, the A'α helix orange, and the Jα helix blue.

Minor things and questions:

The abstract says “with spatiotemporal details that could not be accessible in previous studies” is a vague statement, please explain these previous studies here, and why

→ We appreciate the reviewer's careful reading and detailed comments. To address the reviewer's concern, we have revised the abstract as follows. Please note that we also had to consider the 150-word limit. The parts with major changes are indicated in bold.

“The Light-oxygen-voltage-sensing (LOV) domain superfamily, found in enzymes and signal transduction proteins, plays a crucial role in converting light signals into structural

signals, mediating various biological mechanisms. Although ultrafast spectroscopic studies have revealed the dynamics of the electronic structures of LOV-domain chromophores, understanding the structural changes in the protein moiety, particularly regarding light-induced dimerization, remains challenging. Here, we utilized time-resolved X-ray liquidography to capture the light-induced dimerization of *Avena sativa* LOV2. Our analysis unveiled that dimerization occurs within milliseconds after the unfolding of the A' α and J α helices in the microsecond time range. **Notably, our findings indicate that protein-protein interactions (PPIs) among the β -scaffolds, mediated by helix unfolding, are pivotal in dimerization. In this work, we offer novel structural insights into the dimerization of LOV2 proteins following structural changes in the A' α and J α helices, as well as valuable mechanistic insights into the protein-protein association process driven by PPIs.**"

Page 12line 227, please explain how many candidate structures were investigated

→ As described in the Method section of the original manuscript, we performed the MD simulations for the candidate structures with monomer conformations considering the following two representative conformational frames: (i) a monomer structure with a folded J α helix, and (ii) a monomer structure with an unfolded J α helix. In the MD simulation for (i), 100 ns production simulation was conducted and 10,000 candidate structures with (i) were sampled from the MD trajectory at 10 ps intervals. For (ii), we applied non-equilibrium molecular dynamics (NEMD) simulations to unfold the J α helix along the five directions of $\pm x$, $\pm y$, and z (Supplementary Fig. 10). We used the representative structure from the simulation of (i) as an initial structure for the five NEMD simulations. The NEMD simulations were performed along the hypothetical reaction coordinates until the J α helix completely unfolded, and around 5,000 structures were sampled from each MD trajectory. For the candidate structures with dimer conformations, we performed 100 ns MD simulations considering the six representative conformational frames (Fig. 4a) and sampled 10,000 candidate structures from each MD trajectory at 10 ps intervals. As a result, in the original manuscript, a total of 95,000 candidate structures (35,000 candidates with monomer conformations and 60,000 candidates with dimer conformations) were used during the structural analysis.

In the revised manuscript, we performed additional MD simulations for the six structural frames of the dimer conformations, extending the simulations from the original 100 ns to a total of 300 ns in the revised manuscript. Furthermore, an additional 20,000 candidate structures were sampled from each MD trajectory, on top of those used in the original manuscript. As a result, in the revised manuscript, a total of 215,000 candidate structures (35,000 candidates with monomer conformations and 180,000 candidates with dimer conformations) were used during the structural analysis. We have revised the following sentences in the Methods section to mention the total number of candidates as follows, addressing the reviewer's comment.

"10,000 candidate structures with (i) were sampled from the MD trajectory at 10 ps intervals."

"From each NEMD trajectory, 5,000 candidate structures for (ii) were sampled."

“30,000 candidate structures with the dimer conformations were sampled from each MD trajectory.”

Page 18, the authors conclude that “formation of I₂ should involve the unfolding of the two helices, exposing the β-scaffold to the external environment, which can lead to AsLOV2 dimerization through the dimeric interaction formed between the β-scaffolds of two different monomers”. Is it possible to discuss the strength of this conclusion please. It is difficult to follow some of the reasoning. Which changes are isolated from MD to support this. It would be helpful if the conclusions could more clearly summarise the strength of the evidence. This is important because conclusions arise from a combination of excellent quality experimental results, and MD simulations which are never guaranteed to be correct. Please try to phrase this as realistic as possible, I am suggesting to identify strengths and weakness for the correspondence between MD and experiment.

→ We appreciate the reviewer's careful reading and detailed comments. To clarify the reasoning behind the conclusion that 'formation of I₂ involves the unfolding of the two helices, exposing the β-scaffold to the external environment, which can lead to AsLOV2 dimerization through the dimeric interaction formed between the β-scaffolds of two different monomers.', we have revised the discussion section as follows. Specifically, we moved the sentences from line 371 to line 374 to follow the sentence on line 394, and combined the paragraph starting at line 364 with the paragraph starting at line 375. The revised parts are highlighted in bold.

“Subsequently, I₂ converts into P with a time constant of 10.6 ms for WT. [... omitted ...] In particular, in each monomer that constitutes the optimal structure, the Jα and A'α helices are unfolded, and the β-scaffolds form a dimeric interface (Fig. 4). **Regarding this, in our structural analysis, candidates with the homo L/L dimer conformations exhibit a better fit to SADS₃ when compared to those with hetero L/G dimer conformations. [... omitted ...] For the dimer conformations, docking simulations also indicate that the formation of dimers is more favorable in the homo conformations than in the hetero conformations (Fig. 4a and Supplementary Table 1), aligning with the results of our structural analysis (See Supplementary Note 3, for details). These structural characteristics suggest that the formation of I₂ should involve the unfolding of the two helices, exposing the β-scaffold to the external environment, which can lead to AsLOV2 dimerization through the dimeric interaction formed between the β-scaffolds of two different monomers.**”

We also added the following sentences in the conclusion section.

““[Omitted ...] The results of this structural analysis are also supported by additional SEC experiments and docking simulations, which show that the dimeric interaction of the protein is strengthened in the helix-deleted constructs (ΔJα, ΔA'α, and ΔJα/ΔA'α). [Omitted ...] Although our experimental and theoretical approaches successfully captured the structural changes associated with the formation of the dimer conformation for AsLOV2, additional studies with improved time resolution and SNR, such as those employing serial crystallography, are necessary to elucidate the local structural changes involved in the photoadduct formation between the chromophore and the core domain of the protein, as these changes are considered to be crucial structural components for the light-induced structural transitions. [Omitted ...]”

SI page 13, figure 5. remain data for the light state pure or is it extrapolated? If it's extrapolated then please fully explain the quantification. Has there been made an attempt to fit coordinates to this envelope

→ The scattering data in the light state mentioned by the reviewer originated from the protein sample exposed to 450 nm wavelength for over 30 mins, achieving a saturated light state. This sample was also continually illuminated at 450 nm during the measurement. We note that the data represent direct observations of the light-illuminated sample, not estimations or extrapolations for the light state. Regarding this, we added the following discussion and a new figure (Supplementary Fig. 12) to the revised Supplementary Information.

“Note 2. Structural analysis of the static X-ray scattering curve for *AsLOV2* in the light state using the optimal structures obtained from TRXL

As mentioned in the main text, we utilized the static scattering curve of the light state to perform the reconstruction of the low-resolution structure. We did not conduct the process of refining atomic coordinates that describe the static scattering curve. Instead, as we had already found optimal conformations that well depict the difference scattering curve of the light state during the structural analysis on the TRXL data, we examined whether these optimal structures could accurately describe the scattering curve of the light state. As a result, a linear combination of the theoretical curves calculated by using the optimal structures of G, I₁, I₂, and P obtained from the structural analysis of TRXL data, with the ratio of 1(± 2.8):0(± 2.3):66(± 0.6):33 (± 0.4) for G, I₁, I₂, and P, shows good agreement with the experimental static curve in the small-angle region (0.025 Å⁻¹ - 0.2 Å⁻¹), which is sensitive to global structural properties of proteins such as the oligomer state (Supplementary Fig. 12). This result indicates that under the photo-saturation condition, while most of G converts into I₂, only a portion of I₂ forms P with the dimer conformation. This result is consistent with the observed photoconversion yield of *AsLOV2*, which ranges from 0.1 to 0.3, as observed in our SEC experiment results and previous works (Kawano, F. et al., *PLoS One*. **8**, e82693 (2013); Kennis, J. T. M. et al., *Biochemistry* **42**, 3385-3392 (2003)). Furthermore, this result indicates that under the photo-saturation condition, *AsLOV2* predominantly exists as a mixture of I₂ with the monomer conformation and P with the dimer conformation. The coexistence of these two conformations is consistent with the fact that the R_g value of 22.72 Å, estimated from the static curve of the light state, is smaller than that of 24.76 Å obtained from the dimer conformation of P.”

Supplementary Figure 12. Structural analysis of the static scattering curve (black) of *AsLOV2* in the light state using optimal structures obtained from analysis based on **TRXL data.** The theoretical curve (red) of the light state was calculated as a linear combination of theoretical curves computed using optimal structures of the ground state (G), two intermediate states (I_1 and I_2), and the photoproduct (P), with the ratio of 1:0:66:33 for G, I_1 , I_2 , and P.

We also added the following sentence in the Results section.

“Subsequently, we investigated whether the optimal structures of G, I_1 , I_2 , and P could accurately describe the static scattering curve of the light state. The linear combination of the theoretical curves, calculated using the optimal structures, showed good agreement with the experimental static curve in the small-angle region (See Supplementary Note 2 and Supplementary Fig. 12 for details).”

SI figs 5,6,7. As you know the low- q information is essential in order to properly fit the data and extract the correct contrast. I am surprised the SAXS beamline does not go to lower q range. Can you explain what limited the q range, and how zero q extrapolated amplitudes were estimated

→ We are sorry for the confusion caused by displaying SAXS data to match the TRXL data plotted from 0.045 \AA^{-1} in the old version. In our static scattering data, the lowest q value is around 0.025 \AA^{-1} . This static data for *AsLOV2* was then extrapolated using the indirect fourier transform method of the GNOM program for the reconstruction of low-resolution structures for the dark and light states. We have revised Supplementary Fig. 7 (Supplementary Fig. 5 in the original Supplementary Information) to clearly depict the lowest q value, ensuring it corresponds to approximately 0.025 \AA^{-1}

Supplementary Figure 7. Static SAXS data of AsLOV2. a Static SAXS curves for dark (gray) and light states (purple). [... omitted ...]

As the reviewer pointed out, we agree with the concern regarding the impact of the lack of data at low q values on the results of low-resolution structure reconstruction analysis based on static scattering data. Nevertheless, we concluded that the lack of scattering data in the small q range below $q \sim 0.025 \text{ \AA}^{-1}$ does not have a significant impact on the structural analysis based on the static data. Here is why. The R_g values estimated from the static scattering data suggest that light activation of AsLOV2 involves an increase in the R_g value of the protein, a phenomenon similarly observed in structural analyses based on TRXL. Additionally, the low-resolution structures of the dark and light states reconstructed from the static data were similar to the monomer structure of AsLOV2 in the dark state and the dimer structure of the final photoproduct obtained from our structural analysis based on the TRXL data, respectively. Notably, the elution profiles of the protein in dark and light states obtained from our size-exclusion chromatography (SEC) measurements reveal that the light-induced transition of AsLOV2 involves a dimerization process. This finding corroborates the structural changes from monomer to dimer conformations observed in both the reconstructed structures from the static data and the optimal structures from the TRXL data. Moreover, as mentioned earlier, it was further confirmed that the structural information obtained from both the static curve and TRXL data demonstrates consistency.

Responses to the comments from Reviewer #2

This manuscript reports on a light-activated protein-protein interaction mechanism in the LOV2 domain. The authors use time-resolved X-ray liquidography (TRXL) to monitor first – an unfolding event, which is then followed by a dimerization process. The proposed mechanism is substantiated by mutational studies and chromatography. This manuscript presents very nice TRXL data – and the kinetic modeling and – in particular – the MD-based structural refinement approaches are impressive and contain novel features that can be

transferred to other target systems. However, the biological aspect(s) should be elaborated upon to better highlight how the results reconcile with prior time-resolved studies of LOV proteins, and put the current results into perspective of LOV signaling in general (see more details below).

→ We appreciate the positive evaluation of the reviewer. In the revised version, we have amended the discussion to better address the comparison between previous time-resolved X-ray liquidography studies of LOV domains and our results. Also, we present the biological perspectives of LOV signaling based on our results.

Major concerns:

** The Abstract and the Introduction are written as if this would be THE mechanism for PPIs – which is most likely not the case. I agree that it can be one out of many mechanisms. Therefore, the manuscript would benefit from a rewriting to present the observed results as THE mechanism for activation of this particular LOV protein. Such specification avoids overarching claims – but also strengthens the manuscript and makes it more to the point. And as a consequence, the Title should be revised.*

→ We appreciate the careful reading and the detailed comments from the reviewer. We agree with the reviewer's point that the abstract and introduction of our manuscript may imply that the dimerization process of AsLOV2 is the only mechanism for PPIs. In response to this feedback, we have revised the title, abstract, and introduction of our manuscript to clarify that the structural mechanism of photo-induced dimerization elucidated in our study represents one of the structural templates for understanding protein associations mediated by protein-protein interactions. The parts with major changes are indicated in bold.

First, we changed the title from “Structural dynamics of protein-protein association unveiled by time-resolved X-ray liquidography” to “**Structural dynamics of protein-protein association involved in the light-induced transition of *Avena sativa* LOV2 protein**”.

We also revised the abstract as follows.

“The Light-oxygen-voltage-sensing (LOV) domain superfamily, found in enzymes and signal transduction proteins, plays a crucial role in converting light signals into structural signals, mediating various biological mechanisms. Although ultrafast spectroscopic studies have revealed the dynamics of the electronic structures of LOV-domain chromophores, understanding the structural changes in the protein moiety, particularly regarding light-induced dimerization, remains challenging. Here, we utilized time-resolved X-ray liquidography to capture the light-induced dimerization of *Avena sativa* LOV2. Our analysis unveiled that dimerization occurs within milliseconds after the unfolding of the A'α and Jα helices in the microsecond time range. **Notably, our findings indicate that protein-protein interactions (PPIs) among the β-scaffolds, mediated by helix unfolding, are pivotal in dimerization. In this work, we offer novel structural insights into the dimerization of LOV2 proteins following structural changes in the A'α and Jα helices, as well as valuable mechanistic insights into the protein-protein association process driven by PPIs.**”

The introduction was also revised:

“Photoreceptors detect light signals, a vital environmental stimulus for living organisms, and convert them into biological signals. This enables the organisms to adapt to and respond to changes in their light environment. The LOV superfamily acts as photosensory units in various enzymes and signal transduction proteins with flavin mononucleotide (FMN) as their chromophores. The core structure of LOV domain consists of five central β -sheets (A β , B β , G β , H β , and I β) and four α -helices (Ca, Da, Ea, and Fa), which are highly conserved throughout blue-light receptors (Fig. 1). Upon irradiation, LOV domains utilize FMN to sense blue light, converting such stimuli into structural signals through conformational changes in the protein moiety.

Various studies have been conducted on the photoresponse of these LOV domains. These studies have proposed that, upon activation by blue light, a photoadduct is formed through a covalent bond between FMN and cysteine located within the Ea of the domains, followed by subsequent structural changes in the protein moiety of LOV domains. **Especially, photoreceptors that possess LOV domains as light-sensing regions undergo dimerization during their light-induced signal transduction. These structural changes play a crucial role in modulating downstream biological processes.**

Although spectroscopy studies have suggested that various LOV domains may form transient dimers, the dimerization observed in these studies was dependent on concentration, even without light exposure. Moreover, while ultrafast studies effectively probe the electronic structure of the chromophore, they do not directly detect global structural changes of the proteins involved in photo-induced dimerization. Notably, as summarized in Supplementary Figure 1, several structural studies on the proteins have reported the dimer conformations of LOV domains, suggesting that these proteins can adopt dimer conformations with various dimeric interactions. However, they do not offer detailed structural insights into the mechanisms of the photo-induced dimerization. These limitations have made it challenging to obtain specific structural information about dimerization, leaving the structural characteristics of LOV domain dimers and their dimerization processes elusive.

In this work, to overcome these challenges, we apply time-resolved X-ray liquidography (TRXL) to capture the photoinduced structural dynamics in *AsLOV2*, which does not show concentration dependence on dimerization in the dark state (Fig. 1 and Supplementary Figs. 2a and 2b). *AsLOV2* (residue 413-520) belongs to the LOV2 domain of *Avena sativa* phototropin 1 (*AsPhot1*), sharing the same structural framework as canonical LOV domains. We used a construct that incorporates the A' α helix (residue 404-412) connecting to the LOV1 domain on one side and the J α helix (residue 521-546), which links to the serine/threonine kinase (STK) domain on the other side of *AsLOV2*. Due to the presence of the A' α and J α helices, this construct maintains a monomeric state in the dark, making it a suitable model system for observing transient dimerization upon irradiation. In this work, we refer to this construct including the A' α and J α helices (residue 404-546) as *AsLOV2*. **TRXL, also known as time-resolved X-ray solution scattering (TRXSS), is one of the sensitive experimental methods for detecting global conformational changes in solution induced by a trigger such as optical excitation and temperature jump. When TRXL is applied to macromolecules, it is also referred to as TR small-angle/wide-angle X-ray scattering (TR-**

SAXS/TR-WAXS). Kinetic analysis of the TRXL data from both wild type (WT) and I427V mutant, the latter of which exhibits a relatively faster dark recovery compared to WT, identified three distinct intermediates connected by two time constants on the microseconds and milliseconds timescales in the light-induced structural transition of LOV2. Additionally, structural analysis aided by molecular dynamics (MD) simulations revealed that LOV2 undergoes extensive conformational changes related to the changes in oligomeric state (dimerization) through two monomeric intermediate states. Regarding the dimerization, we demonstrated that the formation of the dimeric interface for *As*LOV2 is associated with PPIs driven by the inter-monomeric interaction between the β -scaffolds, as evidenced by size exclusion chromatography (SEC) experiments on three *As*LOV2 constructs: a construct with the $J\alpha$ helix deleted ($\Delta J\alpha$), a construct with the $A'\alpha$ helix deleted ($\Delta A'\alpha$), and a construct with both $J\alpha$ and $A'\alpha$ helices deleted ($\Delta J\alpha/\Delta A'\alpha$). Moreover, we observed that the SEC profiles of K413E and E475K mutants, designed to disrupt the electrostatic interaction between *As*LOV2 monomers, exhibit a decreased dimer fraction compared to that of the WT. **These observations support that the PPIs, including not only the interaction between the β -scaffolds but also a local electrostatic interaction, contribute to the formation of the dimeric interface. The findings of our study provide direct insights into the structural mechanism of photo-induced dimerization in *As*LOV2, as well as a structural template for understanding the process of protein-protein association driven by PPIs.”**

** TRXL studies have been performed on LOV proteins e.g. Berntsson et al., Structure, 2017 (cited) & Henry et al., Biochemistry, 2020 (not cited). Since the timescales in part overlap with those in the current manuscript and the proteins are (very) closely related – this should be commented upon. E.g. similarities/differences in the protein origins & constructs, dS profiles, and structural interpretations.*

→ We appreciate the reviewer's careful reading and detailed comments. Actually, our original manuscript had citations for both LOV TRXL studies (Berntsson et al., *Structure* **25**, 933-938 (2017); Henry et al., *Biochemistry* **59**, 3206-3215 (2020)) referred to by the reviewer (Refs 47 and 48 in the old version). We did not discuss the details of those studies because our study primarily focuses on the structural changes involved in the light-induced monomer to dimer transition of *As*LOV2, which differs from the light-induced structural changes observed in the target systems of the previous studies. Specifically, Refs 47 and 48 concern structural studies on the light-induced dimer-to-dimer transition of dimeric *Bs*YtvA without the $J\alpha$ helix, and the light-induced monomer-to-monomer transition of monomeric *Cr*LOV1, respectively.

Nevertheless, considering that both the previous studies and our study share similarities in the kinetics for the formation of I_1 , we agree with the reviewer's comment that discussing the similarities and differences between the previous studies and our own will make our manuscript more informative. Therefore, we have further discussed these similarities and differences by adding the following sentences to Discussion.

“Meanwhile, a previous structural study reported that the light-activated LOV domain from *Bacillus subtilis* (*Bs*YtvA) forms I_1 with a time constant of $\sim 2 \mu\text{s}$, which is similar to that observed for I_1 formation of *As*LOV2 in our study. Considering that *Bs*YtvA is a type of LOV protein without the $J\alpha$ helix, this similarity suggests that the $J\alpha$ helix is not significantly

involved in the formation of I₁ during the light-induced transition of LOV proteins. This is consistent with our structural analysis, which shows that structural changes in the A'α helix and β scaffold are involved in the formation of I₁. [...omitted...] Furthermore, previous studies on LOV domains of phototropin proteins, a type of LOV domains found in LOV proteins with an effector domain, have suggested that the light-activated LOV proteins exhibit structural changes in their β-scaffolds (Henry et al., *Biochemistry* **59**, 3206-3215 (2020)), which strengthen their dimeric interactions. These structural changes have implied that the enhanced dimeric interactions, in turn, facilitate the activation of downstream histidine kinase domains by promoting structural rearrangement of the downstream moieties in phototropin proteins (Henry et al., *Biochemistry* **59**, 3206-3215 (2020)). Considering these findings and the fact that the effector domain of AsPhot1 is connected to the C-terminus of its LOV2 domain (AsLOV2), the structural mechanism proposed in our study (Fig. 6) suggests that the light-induced dimerization of AsLOV2 influences downstream reactions by reorganizing the structural arrangement of the effector domains and enhancing the interaction between them in AsPhot1 (Figs. 1a and 6). Our study provides a structural interpretation of AsLOV2's light-induced dimerization, as summarized in Figure 6, which offers a conformational template for understanding how LOV proteins with an effector domain utilize light to regulate the structural and functional properties of their effector domains. The structural mechanism of the dimerization for AsLOV2 suggests that protein oligomerization can be mediated by oligomeric interfaces facilitated by structural changes in subunits.”

** The underlying biology should be more elaborately described. For instance, how general is the mechanism identified in this work given that other LOV activation mechanisms have been presented, e.g. repositioning of LOV domains (Heintz et al., 2016, eLIFE) & monomerization (Conrad et al., 2013, Biochemistry). AND – what are the consequences for downstream signaling – which is not mentioned so much in the current version.*

→ We appreciate the reviewer's careful reading and detailed comments. We have revised Discussion to provide a comprehensive biological perspective on the diverse LOV domains and to offer insights related to downstream signaling based on the AsLOV2 TRXL results, as follows.

“For LOV proteins without an effector domain (short LOV domains), previous structural studies have shown that LOV domains within these proteins exhibit various conformational changes, including dimerization and dissociation, during light-induced transitions (Conrad, K. S. et al., *Biochemistry* **52**, 378-391 (2013); Zoltowski, B. D. et al., *Biochemistry* **47**, 7012-7019 (2008)). These structural changes enable the LOV proteins to interact with their partner proteins or molecules and mediate signal transduction processes. In contrast, LOV domains in LOV proteins with an effector domain exhibit structural changes that are synchronized with the modulation of the effector domains (Heintz, U. et al., *Elife* **5**, e11860 (2016); Takakado, A. et al., *Phys. Chem. Chem. Phys.* **19**, 24855-24865 (2017)). For several LOV proteins with an effector domain, it has been observed that the β-scaffolds in their LOV domains form the primary dimerization interface, implying a shared structural characteristic among LOV domains concerning their dimeric interfaces. [...omitted...].”

Minor concerns:

** Overall this is a well written manuscript, but a careful readthrough will likely correct a few phrasings/repetitions here and there throughout the manuscript.*

→ We appreciate the positive evaluation of the reviewer. We revised the manuscript to make it more accurate and concise.

** Sometimes “microsecond” is used and sometimes “ μ s” – same for “milliseconds” vs “ms” I believe.*

→ We appreciate the careful reading of the reviewer. In the revised manuscript, except for the first mention of microsecond and millisecond, we revised them to μ s and ms, respectively.

** p. 4, l. 64 – I427V mutant is mentioned without citation – was this known before or somehow from this study? I guess that a citation is missing.*

→ We thank the reviewer for spotting this. As the reviewer commented, the I427V mutant of AsLOV2 is well-known for exhibiting a faster photocycle compared to that of WT. We added the citation for the corresponding paper in the revised version (Christie, J. M. et al. *Biochemistry* **46**, 9310-9319 (2007)).

** The Results start with the mutant data – which we learn later on (end of Results section) is because of better signal-to-noise ratio. This was confusing to me. Can WT and mutant be presented side-by-side? Then it even be more apparent that they are similar and still be logical.*

→ The comparison of TRXL data between WT and I427V mutant was already presented side-by-side in Supplementary Fig. 6 in the previous version (Supplementary Fig. 3 in the revised version). However, to clearly present the TRXL results for WT and I427V in the main text, we added detailed descriptions about the reason for measuring I427V mutant TRXL data in the first paragraph of the result section.

We revised the result section as follows. The parts with major changes are indicated in bold.

From:

“To investigate the light-induced structural changes of AsLOV2, we measured TRXL data from microseconds (μ s) to hundreds of milliseconds (ms) using the laser pump-X-ray probe scheme³⁸⁻⁴¹ (details in Methods). The TRXL data were collected from 5.62 μ s to 316 ms for WT and from 5.62 μ s to 100 ms for the I427V mutant.”

To:

“To investigate the light-induced structural changes of AsLOV2, we measured TRXL data from microseconds (μ s) to hundreds of milliseconds (ms) using the laser pump-X-ray probe scheme³⁸⁻⁴¹ (details in Methods). **TRXL experiments were conducted for both WT and I427V mutant. The I427V mutant⁵⁶, which exhibits a faster photocycle than WT, was deliberately chosen to efficiently collect TRXL data with a higher signal-to-noise ratio (SNR), as the slow recovery of WT required the lowering of the repetition rate for**

measuring data at long time delays. The TRXL data were collected from 5.62 μ s to 316 ms for WT and from 5.62 μ s to 100 ms for the I427V mutant (**Supplementary Fig. 3**).”

** Obvious to some but perhaps it should be mentioned somewhere that the P population sums to 0.5 due to dimerization?*

→ We appreciate the careful reading of the reviewer. We have added a description of the dimerization fraction for P in Fig. 2c (time-dependent population map) in the figure caption as follows.

“Fig. 2. [...omitted...] **Since the protein exhibits dimer conformation in the P state, the maximum population of P is 0.5. This value is half those of the other states (G, I₁, and I₂), which exhibit monomer conformations.**”

** Treatment of the TRXL data (SVD & kinetic modeling) is excellent and the TRXL ability to scrutinize between sequential and parallel models is impressive.*

→ We appreciate the positive evaluation of the reviewer.

** MD-based unfolding of J α helix to model SADS₂ is interesting. However, it would be even more convincing if the authors could show that unfolding a helix somewhat corresponding in size (e.g. F α) would result in worse fits to the data. I.e. a repeat of Suppl. Fig. 8 but for helix F α .*

→ We appreciate the thoughtful comment from the reviewer. In our structural analysis, we obtained candidate structural frames for the light-induced transition of AsLOV2 by classifying the structural information of previously reported LOV domains. During this process, we were able to anticipate that the transition could involve structural changes in the A' α and J α helices, particularly including the unfolding of the J α helix. These structural changes of the helices led us to apply the structural frames that incorporated the structural changes in the A' α and J α helices of the protein in our structural analysis of the monomer-to-monomer transitions of the protein, including the formations of I₁ and I₂. In this context, the unfolding of the F α helix was not included in the scope of our structural analysis. Nevertheless, we agree with the reviewer's suggestion that investigating whether the unfolding of the F α helix which is similar in size to the J α helix adequately or inadequately describes the experimental data for I₂ (SADS₂) would further strengthen our findings. Therefore, we applied MD simulation-aided structural analysis to the F α helix, similar to what was done with the J α helix. As a result, the theoretical difference scattering curves from the candidate structures with the unfolded F α helix did not describe SADS₂, suggesting that the unfolding of the F α helix does not significantly affect the formation of I₂ (Supplementary Fig. 11). To incorporate this additional structural analysis into our manuscript, we have added the following paragraph and Supplementary Figure 11 to Supplementary Information.

“Note 2. The impact of the F α helix unfolding on the formation of I₂.

To further confirm that the unfolding of the A' α and J α helices is the primary structural change driving the formation of I₂, we performed an additional control analysis as follows. AsLOV2 contains four α -helices in its core structure, in addition to the A' α and J α helices (Fig. 1b).

Notably, among these core helices, the F α helix is relatively similar in size to the J α helix. As a control, we further investigated the impact of the F α helix unfolding on the formation of I₂ using the same methods applied in our structural analysis. A brief explanation of this additional structural analysis is as follows.

First, we generated candidate structures with unfolded F α helix using NEMD simulations with the pulling algorithm of GROMACS in a manner similar to those applied to the candidates with unfolded J α helix. For the NEMD simulations, we used a representative structure from the MD trajectory of the monomer structure with a folded J α helix as the initial structure. In the simulations, we defined two residues, Arg460 (Q3) and Asn472 (Q4), representing the N and C-terminus of the F α helix. We then employed the external pulling force such that the Q4 position was moved away from Q3, triggering the unfolding of the F α helix. Specifically, we applied the force to pull the F α helix along the directions of $\pm x$, $+y$, and $+z$ (Supplementary Fig. 11a). All NEMD simulations were performed using the pulling algorithm along the hypothetical reaction coordinates until the F α helix completely unfolded. From each MD trajectory, approximately 1,000 candidate structures for a monomer structure with an unfolded F α helix were sampled. Subsequently, theoretical difference scattering curves were calculated using these candidate structures and compared with SADS₂.

As shown in Supplementary Figure 11f, the theoretical difference curves from the candidate structures with an unfolded F α helix fail to describe the SADS₂, implying that the unfolding of the F α helix does not significantly contribute to the formation of I₂. Furthermore, this finding supports that the primary structural changes driving the formation of I₂ are predominantly due to the unfolding of the A' α and J α helices.”

We also added the following sentences to the “Structural analysis of difference X-ray scattering curves using MD simulations” in the Results section.

[... omitted ...] To further confirm that the formation of I₂ is primarily mediated by the unfolding of the A' α and J α helices, we conducted an additional control analysis. AsLOV2 has a core structure comprising four α -helices, along with the A' α and J α helices (Fig. 1b). Notably, the F α helix is relatively similar in size to the J α helix among these core helices. During the control analysis, we investigated the impact of the F α helix unfolding on the formation I₂, with details described in Supplementary Note 1. The theoretical difference curves from the unfolded F α helix structures failed to describe SADS₂, indicating that the F α helix unfolding does not significantly contribute to the formation of I₂ (Supplementary Note 1 and Supplementary Fig. 11). This supports that the unfolding of the A' α and J α helices is the primary structural change driving the formation of I₂.”

Supplementary Figure 11. Additional structural analysis of SADS₂ (corresponding to I₂) using candidate structures with unfolded F α helix. **a** A schematic diagram illustrating five directions ($\pm x$, $+y$, and $+z$) for the pulling of the F α helix. This additional structural analysis was conducted using the structures obtained through NEMD simulations where the F α helix was pulled in a specific direction ($\pm x$, $+y$ or $+z$). **b** Examples of the simulated structures with the F α helix unfolded in the $+x$ direction. **c** Examples of the simulated structures with the F α helix unfolded in the $-x$ direction. **d** Examples of the simulated structures with the F α helix unfolded in the $+y$ direction. **e** Examples of the simulated structures with the F α helix unfolded in the $+z$ direction. The structure of the F α helix is shown in blue, while the rest of the structure is shown in gray (a–e). **f** Comparison of SADS₂ (black) with the 100 best-fitted theoretical curves from the candidate structures with an unfolded F α helix, which failed to accurately describe the experimental data (gray).

** 100 ns of MD simulations is very short on the biological time scale of dimerization. Therefore, while one E/E dimer with b-frame reproduced the data best, I wonder if any of the remaining models could have achieved an equally good fit given enough simulation time? Perhaps this can be answered by plotting RMSD over time. Seeing that the E/E dimer with b-frame has low RMSD throughout will indicate stability – and perhaps the other systems drift apart. In any case, RMSD vs time plots would be a nice addition.*

→ We conducted structural analysis aided by MD simulations to find the structures that best describe the TRXL data, thereby providing structural insights into the light-induced transition

of AsLOV2. The purpose of our MD simulations was to generate a range of candidate structures with various conformational characteristics. During the simulations, we aimed to establish a candidate structural pool that provides sufficient structural diversity necessary for the analysis, rather than to directly provide structural insights into the light-induced transition. From the MD simulations, we obtained a candidate structural pool that includes a wide range of structural characteristics as described in Fig. 5. This structural diversity enabled us to select the structures that most accurately describe the TRXL data during the subsequent χ^2 test. In particular, in the structural landscapes, the range of structural parameters of the total structural pool is sufficiently broader than those of the optimal structures as shown in Fig. 5, indicating that the MD simulations provide a sufficient level of structural diversity for the structural analysis.

Nevertheless, to address the reviewer's comment, we performed additional MD simulations for the six structural frames of the dimer conformation, extending the simulations from the original 100 ns to a total of 300 ns in the revised manuscript. Subsequently, we extracted an RMSD plot over time from each of the six MD trajectories. For calculating the RMSD plots, the starting structure of the MD simulation for the b-frame, which was selected as an optimal structural frame for the photoproduct, was used as the reference structure. In all six RMSD plots, we observed that the RMSD values had converged around 200 ns. Among them, the RMSD plot for E/E dimer with b-frame (L/L dimer with b-frame in the revised version), selected as the most optimal dimeric conformation during our structural analysis, generally exhibited lower RMSD values compared to those for the other dimer conformations (Fig. R1).

Furthermore, we conducted additional structural analyses to determine whether the extended MD simulations would affect our original results. During the structural analysis, we sampled candidates of dimer conformations at 10 ps-intervals from those six extended MD simulations, and these dimer conformations were added to our original structural pool. These dimer conformations were then added to our original structural pool. Consequently, this updated structural pool includes candidates from both the original structural pool and the extended MD simulations. Following this, we conducted the structural analysis detailed in our manuscript and successfully extracted the optimal structures of the product from this updated pool. As shown in the revised Fig. 4a, the optimal structures still show that the photo-product adopts a homo L/L dimer configuration (light state monomer + light state monomer) with a structural frame where the β -scaffolds interact with each other (β -scaffold– β -scaffold interaction). Moreover, the comparison of the optimal structure from the original structural pool and that from the updated structural pool shows that the two optimal structures exhibit almost identical structural characteristics, except for minor differences in the arrangement of the unfolded $J\alpha$ helix (Fig. R2). This similarity of the two optimal structures indicates that the extended MD simulations did not have a significant impact on our structural analysis. Regarding the use of the updated structural pool, we have added the following sentence in the section on MD simulations-aided structural analysis of TRXL data within the method and have revised Figure 4a.

“During these MD simulations, production simulations were conducted over a period of 300 ns.”

Additionally, we calculated the structural landscapes using the results from the updated structural pool and revised Figure 5 (Fig. 5). Compared to the landscapes in the original manuscript (Fig. R3), the landscapes in the revised version (Fig. 5) include relatively greater diversity in the structural pool. Despite this, they exhibit nearly identical structural changes involved in the light-induced transition of *As*LOV2 as observed in the original version.

Fig. R1. RMSD (Root Mean Square Deviation) plots extracted from 300-ns MD simulation trajectories of six dimer conformation candidates. (a–c) RMSD plots for three hetero L/G dimer conformations with a'–c' frames. (d–f) RMSD plots for three homo L/L dimer conformations with a–c frames. Each RMSD plot includes an inset showing the schematic of the corresponding frame.

Fig. 4. Structural analysis of SADS₃. **a** Schematic diagram showing six representative AsLOV2 dimer conformation candidates and the corresponding residuals between SADS₃ and the best-fit scattering curve determined from the structural analysis. In the upper panel, the

results of structural analysis for three hetero L/G dimers are shown, and in the lower panel, those for three homo L/L dimers are shown. In the schematic of the frames for the homo- and hetero-dimers, the light state LOV2 core domain (L) is depicted in bright yellow, and the non-activated LOV2 core domain (G) is depicted in dark yellow. J α helices are shown in blue, and A' α helices are shown in red. The best-fit curve with the smallest residual and χ^2 value is highlighted with the red box. The docking score (DS) and confidence score (CS) from molecular docking simulations, which assess the plausibility of dimer formation, are shown for each conformation. [...omitted...]

Fig. R2. Comparison of the optimal structure of the P state from 100 ns (semi-transparent gray) and 300 ns (red) MD simulation-aided structural analysis.

Fig. 5. Landscapes of structural parameters in the *As*LOV2 photocycle revealed by the TRXL study. [omitted...]

Fig. R3. Landscapes of structural parameters in the *AsLOV2* photocycle revealed by the TRXL study in the original manuscript. **a** The landscape of R_g as a function of D_{max} . **b** The landscape of distance between Ala523 and Ala543 (D_1) as a function of distance between Phe403 and Glu412 (D_2). **c** The landscape of solvent accessible surface area (SASA) among the A'α, Jα helices, and β-scaffold, as a function of root mean square deviation (RMSD) of the β-scaffold. **d** The landscape of SASA of area 1 as a function of the SASA among the β-scaffold and the remaining structural components of *AsLOV2* excluding the A'α, Jα helices, and β-scaffold. The values were calculated from the optimal structure of the G state (black dots), and these values were compared with those of I₁, I₂, and P. The parameters for I₁, I₂, monomer A, and monomer B are denoted by green, blue, red and magenta dots, respectively. The parameters calculated from the entire structural pool used in the structural analysis are also represented in the landscapes (gray dots).

** Is Fig. 4d referred to somewhere?*

→ We thank the reviewer for spotting this, which is now added in the revised version. The parts with major changes are indicated in bold.

“Moreover, it was predicted that electrostatic interaction between K413 near A'α helix and E475 in the β-scaffold is also related to the formation of the dimeric interface from the

theoretical approach based on PDBePISA4, an effective tool for identifying interactions at protein interfaces (See **Fig. 4d** and Supplementary Methods for details).”

** p. 23 – how were the heating signals removed from the TRXL?*

→ We express our appreciation for the reviewer's careful consideration. To extract only the contributions associated with the structural changes of the target protein system, we reconstructed the solvent heating response as a linear combination of the X-ray scattering change resulting from the temperature change at a constant pressure and from the density change at a constant temperature. This contribution of heating response was then subtracted from the TRXL data. As a result, only the contribution of the structural changes of the protein is extracted. To incorporate an explanation of this process into our manuscript, we added the following paragraph and Supplementary Figure 15 to the Supplementary Methods in Supplementary Information.

“Elimination of the solvent heating contribution from TRXL data

Time-resolved difference scattering curves contain the contributions of structural changes in protein as well as from the heating of the solvent due to energy transfer by excited protein molecules. Using a well-established method, we reconstructed the contribution of the solvent heating response as a linear combination of the X-ray scattering change resulting from the temperature change at a constant pressure and that from the density change at a constant temperature. This heating contribution was then removed from the TRXL data to extract only the contributions related to the structural changes of the target protein system (Supplementary Fig. 15). By subtracting the appropriately scaled solvent heating signal, heating-free difference scattering curves were obtained for both WT and I427V of AsLOV2 at the positive time delays.”

Supplementary Figure 15. Comparison of a TRXL of AsLOV2 with a solvent heating signal. The difference scattering curve of I427V at 56.2 μs time delay (black) is compared with the solvent heating signal (red). The solvent heating signal does not show any significant features in the q -range smaller than 1.0 \AA^{-1} .

We also added the following sentence to “Time-resolved X-ray liquidography (TRXL) experiment and data processing” in the Methods section.

“From the TRXL data, the contribution from the heating of the solvent due to energy transfer by excited protein molecules was removed using a well-established protocol (See Supplementary Methods and Supplementary Fig. 15 for the details).”

** p. 25 – Gromacs citation missing.*

→ We thank the reviewer for spotting this, which is now added in the revised version (D. van der Spoel, et al., *J. Comp. Chem.*, **26**, 1701–1718 (2005)).

** p. 26 – which clustering method?*

→ We performed the clustering process using a method based on the GROMOS algorithm (Daura, X. et al., *Angew. Chem. Int. Ed.* **38**, 236-240 (1999)). To incorporate this information into our manuscript, we revised the sentences in lines 548-550 of our manuscript as follows.

“From the MD trajectories, the representative structure for (i) was selected by using the clustering method based on the GROMOS algorithm and then used as the starting structure for the MD simulation of (ii).”

** p. 26 – were all CRY SOL parameters default?*

→ As the reviewer mentioned, in our structural analysis, the theoretical scattering curves of the candidate structures were calculated using the default parameters of CRY SOL. We added the descriptions of CRY SOL parameters in Methods as follows.

“We calculated the theoretical scattering curves of the candidate structures using the CRY SOL **with all default parameters.**”

** References – too many general references on PPI’s 1-8 – maybe one or two is enough?*

Also, see major concern above about the focus of the manuscript.

→ In response to the earlier reviewer's comment, we revised the introduction to focus on the LOV protein, rather than the protein-protein interaction (PPI). Therefore, we reduced the citations related to PPI, leaving only two references (refs 1 and 4 in the old version) necessary for describing the PPI.

** Suppl. Fig. 1 – why not use the same concentrations in the SEC as in the SAXS experiments?*

→ We appreciate the reviewer for spotting this. We added SEC measurement at concentrations of 1.5 mM and 0.2 mM in the revised version. We confirmed, through SAXS analysis at a concentration of 1.5 mM, that AsLOV2 exists in a monomeric conformation in its dark state, as also corroborated by TRXL analysis conducted at the same concentration in the dark state.

Additionally, we verified no concentration dependence on AsLOV2 by observing similar SAXS curves across various concentration conditions (1.5 mM, 0.5 mM, and 0.2 mM), particularly evident in the low q region (Supplementary Fig. 1a in the previous version), which closely correlates with the trend of oligomerization. Therefore, in the previous version, we did not conduct SEC measurement at the highest concentration (1.5 mM), reasoning that lower concentrations (0.5 mM, 0.1 mM, 0.02 mM) adequately reflect the conditions of the 1.5 mM SAXS experiments. Nevertheless, for the sake of consistency, we added SEC data for the 1.5 mM concentration in the revised version (Supplementary Fig. 2b).

Supplementary Figure 2. [...omitted...] **b** SEC elution profiles at the concentrations of 1.5 mM (blue), 0.5 mM (green), 0.2 mM (yellow), and 0.02 mM (light orange).

Responses to the comments from Reviewer #3

The manuscript provides a detailed insight into the light-induced structural changes of a phototropin LOV domain, which is active in plant photoreceptors and in optogenetic tools. A series of time-resolved SAXS and WAXS recordings are analyzed by integrating theoretical scattering curves from model-based molecular dynamics simulations. The analysis covers the time region of microseconds and milliseconds after which the photochemistry is concluded and the protein responses take place. The LOV domain is one of the most intensively investigated photoreceptor domains because of the small size, easy production and large conformational changes. Accordingly, many studies by time-resolved transient grating and infrared spectroscopy are available and summarized in Tables 1 and 2. The key advance of this study is the spatial resolution achieved by the combination of scattering experiments and MD simulations, in which different model assumptions for the protein were iteratively tested for an

agreement with the experiment. The experiments were well conducted and analyzed and represent an impressive methodological achievement.

→ We appreciate the positive evaluation of the reviewer.

However, novel insight into LOV domain structure and dimerization is limited compared to previous knowledge. The same method has been applied to LOV domains previously (refs 47 to 49) albeit without the present focus on light-induced dimerization. In conclusion, the study nicely summarizes and confirms previous insight into the mechanism of phototropin-LOV dimerization by other techniques but the specific additional insight is unclear.

→ We respectfully disagree with the reviewer's perspective regarding the novelty of our work. Our study provides new structural insights into the light-induced dimerization of AsLOV2, which has not been elucidated in previous studies. The optimal structures from our study offer direct mechanistic insights into how AsLOV2, which adopts a monomer conformation in the dark state, forms a dimer conformation upon illumination through the unfolding of the A' α and J α helices and the structural rearrangement of the β -scaffold. In contrast, the studies mentioned by the reviewer, Refs 47 to 49, did not provide direct structural information on the light-induced dimerization of LOV proteins. The study in Ref 47 (Henry, L. et al. *Biochemistry* **59**, 3206-3215 (2020)) successfully elucidated the structural characteristics of the light-induced "monomer to monomer" (not monomer to dimer) transition in the LOV1 protein from *C. reinhardtii* phototropin. The studies in Refs 48 and 49 (Berntsson, O. et al. *Structure* **25**, 933-938.e933 (2017); Berntsson, O. et al. *Nat. Commun.* **8**, 284 (2017)) provided structural insights into the light-induced "dimer to dimer" (not monomer to dimer) transitions of the YtvA LOV domain and LOV-YF1 chimera protein, respectively. In summary, Refs 47 to 49 did not provide direct structural information on the light-induced dimerization of LOV proteins.

1. Figure 3 represents a central figure of this manuscript and points to a single-residue specific spatial resolution. How realistic is such a resolution from analyzing SAXS/WAXS recordings? In other words: How large is the resolvable change in distance by illumination and how large is the respective error?

→ In our study, the X-ray scattering experiments and associated analytical approaches we employed are expected to identify structural changes related to the light-induced transition of AsLOV2 with errors smaller than sub-Å levels (*Sci. Adv.* **8**, eabm6278 (2022); *Cell Rep. Phys. Sci.*, **2**, 100512 (2021); *Proc. Natl. Acad. Sci.* **117**, 14996–15005 (2020); *J. Phys. Chem. Lett.* **8**, 4413–4418 (2017); *Sci. Adv.* **7**, eabi5514 (2021)). This expectation is supported by the fact that the structural landscape L3 captured sub-Å level changes in the β -scaffold during the transition from I₁ to P via I₂ by comparing the values of RMSD of the β -scaffold (Fig. 5b in the original manuscript).

The static X-ray scattering and TRXL profiles in the SAXS/WAXS regions provide valuable structural information, including molecular shape, structural complexity, radius of gyration (R_g), and reconstructed low-resolution structures, for target systems. However, these scattering profiles themselves face challenges in providing single-residue specific spatial resolution. Nevertheless, with appropriate computational modeling, those profiles can offer structural information at atomic resolution. In our study, to overcome the limitation, we applied

MD simulation-aided structural analysis to the TRXL data. In the structural analysis, a pool of candidate structures, including various target conformational frames, was generated from the MD simulations. These candidates were then used to calculate theoretical curves. By identifying which of these curves accurately described the TRXL data, we were able to determine the 10 best structures of the two intermediates and the photoproduct involved in the light-induced transition of AsLOV2. Furthermore, by comparing these structures, we successfully obtained structural insights at atomic resolution into the conformational changes associated with the transition.

*2. Key additional insight is obtained by studying truncated constructs of LOV without extending helices by SEC. The SEC peaks in Supp Figure 10 demonstrate some dimerization of LOV under illumination. The ratio of dimer to monomer is very low (15%) and at odds with time-resolved SAXS. Other studies on LOV have demonstrated a nearly 100% shift from monomer to dimer or to a dynamic equilibrium in the SEC profile (see for example Zoltowski and Crane 2008 *Biochemistry* 47 7012). At which wavelength was the detection of the SEC profiles performed? The scattering profiles seem to provide more solid evidence for the dimerization than the SEC. Accordingly it would be helpful to study truncated constructs by SAXS instead of SEC.*

→ As summarized in Table R1, photoconversion yields for LOV proteins from various origins have been reported in previous studies (Kennis, J. T. M. et al., *Biochemistry* **42**, 3385-3392 (2003); Kawano, F. et al., *PLoS One* **8**, e82693 (2013); Song, S. H. et al., *Photochem. Photobiol. B.* **81**, 55-65 (2005); Kennis, J. T. M. et al., *J. Am. Chem. Soc.* **126**, 4512-4513 (2004); Bannister, S. et al., *Struct. Dyn.* **6**, 034701 (2019)). Table R1 shows that LOV proteins exhibit photoconversion yields ranging from several percent to tens of percent. This suggests that the proteins may display varying photoconversion yields influenced by multiple factors, including whether they are LOV1 or LOV2, as well as other characteristics associated with their origins. In particular, a spectroscopic study reported that the fluorescence quantum yield of AsLOV2 is approximately 13%, indicating that AsLOV2 intrinsically exhibits a low photoconversion yield. It is speculated that this low yield resulted in the observation of light-induced dimerization with a low yield in the SEC profile of AsLOV2 in the light state.

Table R1. Photoconversion yield of various LOV proteins.

	AsPhot1- LOV2 ^{1,2}	CrPhot- LOV1 ³	CrPhot- LOV2 ³	Phy3- LOV ⁴	PtAUREO1c ⁵	PtAUREO1a ⁵
Photoconversion yield (%)	13	17	12	20 ~ 25	23	64

¹*Biochemistry* **42**, 3385 (2003).

²*PLoS One* **8**, e82693 (2013).

³*J. Photochem. Photobiol. B.* **81**, 55-65 (2005).

⁴*J. Am. Chem. Soc.* **126**, 4512-4513 (2004).

Regarding the reviewer's comment, we estimated the yield of AsLOV2 using the SAXS profile of the light state. To do this, we examined whether the optimal structures from structural analysis based on TRXL data could accurately describe the SAXS profile. As a result, we confirmed that the linear combination of the theoretical static scattering curves from the optimal structures with monomer and dimer conformations effectively describes the SAXS profile (Supplementary Fig. 12 in the revised version). Through this process, we were able to estimate the ratio of monomer to dimer from the SAXS profile as 66:33 (Supplementary Fig. 12). We found that the photoconversion yield of the dimerization process under the experimental condition of the SAXS measurement was about 0.3, which is higher than that observed in our SEC experiments.

Supplementary Figure 12. Structural analysis of the static scattering curve (black) of AsLOV2 in the light state using optimal structures obtained from analysis based on TRXL data. The theoretical curve (red) of the light state was calculated as a linear combination of theoretical curves computed using optimal structures of the ground state (G), two intermediate states (I₁ and I₂), and the photoproduct (P), with the ratio of 1:0:66:33 for G, I₁, I₂, and P.

It is speculated that the observed difference in photoconversion yield may arise from the varying conditions applied in the SAXS and SEC experiments. Considering the low photoconversion yield of AsLOV2, we measured the SAXS profile of the light state under the photo-saturated condition to predominantly capture signals from the light state. For this condition, the protein sample was exposed to 460 nm wavelength for over 30 minutes prior to the measurement of the SAXS profile. Furthermore, the sample was continually illuminated at the same wavelength during the measurement. On the other hand, the samples used for SEC measurements, which had been exposed to 460 nm light for over 30 minutes prior to the

experiment, could not be continuously illuminated at the same wavelength during the measurements of the light states due to limitations in the experimental setup. This difference in the experimental approaches is thought to be the reason for the varying photoconversion yields observed between the SAXS and SEC experiments.

Nevertheless, the purpose of our SEC and SAXS experiments was not to determine the photoconversion yield of AsLOV2. Instead, we aimed to identify which conformers are involved in the ground and light states of AsLOV2 using the SEC profiles, and to obtain structural information about these states using SAXS. Our SEC results successfully demonstrated that light-induced dimerization is involved in the photoactivation of AsLOV2. Consistent with these SEC results, the low-resolution structures reconstructed from our SAXS profiles exhibited shapes similar to the monomer and dimer conformations of the protein in its ground and light states, respectively.

Additionally, as shown in the simulated curves in Figure R4, the difference scattering curve for the monomer-to-monomer transition exhibits significantly lower scattering intensity compared to that of the monomer-to-dimer transition. Based on this observation, we conclude that the structural information related to the formation of the photoproduct, contained in SADS₃, can be adequately described solely by the transition from monomer to dimer.

Figure R4. Comparison of simulated curves for monomer-to-monomer transitions and monomer-to-dimer transition. The simulated difference curve for the monomer-to-monomer transition without the $J\alpha$ helix was obtained by subtracting the optimal structure for the ground state (G) from that for I_1 . Similarly, the difference curve for the monomer-to-monomer transition with the $J\alpha$ helix was obtained by subtracting the optimal structure for G from that for I_2 . Lastly, the difference curve for the monomer-to-dimer transition with the $J\alpha$ helix was achieved by subtracting the optimal structure for G from that for P.

In the biochemical field, it is widely recognized that SEC is the most suitable tool for detecting changes in oligomeric states of target proteins, even though it does not provide direct structural details of target proteins. The specificity of SEC in separating molecules based on their oligomeric state, where the position of the elution peak directly corresponds to the state, makes it exceptionally precise and sensitive for detecting the changes in the oligomeric state. In the case of SAXS, the SAXS profile can provide direct information on the global structure of proteins due to the inherent sensitivity of X-rays to the structure of target molecules. In this regard, if the target samples are homogeneous in the solution, SAXS can provide information about changes in oligomeric states as well as the associated conformational changes. However, if the samples exhibit heterogeneous conformational properties in solution, SAXS encounters difficulties in detecting changes in their oligomeric states as directly and clearly as SEC, because the structural sensitivity of X-ray results in scattering signals from heterogeneous conformations mixed together in the SAXS profile.

Nevertheless, to investigate the effect of deleting the $J\alpha$ helix on the dimerization of AsLOV2 using SAXS, we conducted additional experiments on the following four AsLOV2 constructs: (i) WT, (ii) a construct with the 15 residues deleted from the C-terminal of the $J\alpha$ helix ($\Delta J\alpha 15$), (iii) a construct with the 20 residues deleted from the C-terminal of the $J\alpha$ helix ($\Delta J\alpha 20$), (iv) a construct with the $J\alpha$ helix fully deleted ($\Delta J\alpha$). As shown in Figure R5, we successfully measured the SAXS signals for WT and $\Delta J\alpha 15$. However, $\Delta J\alpha 20$ and $\Delta J\alpha$ encountered stability issues, leading to sample aggregation during transport to the synchrotron facility, which made it impossible to measure their SAXS signals.

The SAXS signals for WT and $\Delta J\alpha 15$ in the dark state were measured in the q -range of $\sim 0.0088 \text{ \AA}^{-1} < q < \sim 0.188 \text{ \AA}^{-1}$ at Pohang Light Source II (PLS-II) (Fig. R5). The comparison of the signals shows significant differences in the small-angle region ($q < \sim 0.2 \text{ \AA}^{-1}$), which is sensitive to the global structural properties of proteins, indicating that WT and $\Delta J\alpha 15$ have distinct global structural characteristics (Fig. R5a). Furthermore, the comparison of radius of gyration (R_g) values for WT ($\sim 16.35 \text{ \AA}$) and $\Delta J\alpha 15$ ($\sim 21.23 \text{ \AA}$), approximated from the SAXS data, shows that $\Delta J\alpha 15$ has a relatively larger R_g value compared to WT. Notably, the R_g value of $\sim 21.23 \text{ \AA}$ for $\Delta J\alpha 15$ lies between the R_g value of $\sim 16.35 \text{ \AA}$ for WT and the R_g value of $\sim 24.76 \text{ \AA}$ for the optimal structure of the photoproduct (P) with dimer conformation from the structural analysis of TRXL data (Fig. R5b). Considering the SEC profile of WT, which shows that the construct predominantly adopts a monomer conformation in the dark state, the comparison of the R_g values suggests that $\Delta J\alpha 15$ exists as a mixture of monomers and dimers in the dark state. These results from the SAXS experiment are consistent with the results from the SEC measurements for $\Delta J\alpha$, which demonstrate that the removal of the $J\alpha$ helix can lead to AsLOV2 forming a dimer conformation even in the dark state.

Figure R5. Static SAXS data of WT and $\Delta J\alpha 15$ in the dark state. **a** Static SAXS curve for dark (black) and the construct with the 15 residues deleted from the C-terminal of the $J\alpha$ helix ($\Delta J\alpha 15$) (red). **b** Radius of gyration (R_g) values for WT and $\Delta J\alpha 15$ in the dark state obtained from the SAXS data, and R_g value for P obtained from the optimal structure of P.

3. Figure 6. The depicted structural model of the final P state is puzzling because it infers a strongly extended and flexible structure of the helices which is not in agreement with the constraints from SAXS (Fig. 4c). D_{max} values should limit the extension of the dimer by defining the maximal average distance of scatter centers in the ensemble. The real structure should be an envelope of snapshots showing the flexibility of the helices in solution. Moreover, the chosen representative structures show a residual secondary structure in both helices. Is this residual helical structure only a snapshot in an otherwise unfolded scenario or a representative picture of the P state?

→ We appreciate the reviewer's thoughtful concern. We agree with the reviewer's comment that AsLOV2 exhibits flexible structural characteristics. The optimal structure of P is a representative structural model of the dimer conformation for AsLOV2 in the final photoproduct state, showing that the protein adopts the b-frame dimer conformation with unfolded $A'\alpha$ and $J\alpha$ helices in the final state as shown in Fig. 4a. This structure does not imply that the protein adopts this specific, single conformation exclusively in the state. We also

expect that, in addition to P, the two intermediates will exhibit structurally flexible characteristics influenced by their secondary structures, specifically A'α and Jα helices. Therefore, during the structural analysis based on TRXL data, for each state (I₁, I₂, or P), we extracted the 10 best structures that showed the best agreement with the experimental data. Among those 10 structures, the optimal structure showing the highest agreement with the experimental data was selected as the representative structure for each state. The optimal structures of I₁, I₂, and P were then used to compare the structural characteristics of the states. Furthermore, the dynamical properties of I₁, I₂, and P were already investigated by constructing the structural landscapes depicted in Fig. 5. For the construction of the landscapes, various structural parameters were calculated by using the representative structure of G, the 10 best structures for the light-activated states (I₁, I₂, and P), and total candidate structural pool as described in “Structural analysis of difference X-ray scattering curves using MD simulations” of the Results section. From the landscape, we elucidated the detailed structural information involved in the light-induced dimerization of *AsLOV2*. Furthermore, for each light-activated state (I₁, I₂, or P), the 10 best structures exhibit fluctuations for each structural parameter (R_g , D_{max} , D1, D2, SASA of area 1, SASA of area 2, and RMSD of the β scaffold), indicating that *AsLOV2* exhibits dynamic structural characteristics in its light-activated states (Fig. 5). The comparison of the 10 best structures of P shows that these structures exhibit significant structural differences, particularly within the unfolded helices (Fig. R6). These differences further demonstrate that *AsLOV2* exhibits structural flexibility in the P state. To clarify this aspect in the manuscript, we revised the results, discussion, and methods sections.

“Structural analysis of difference X-ray scattering curves using MD simulations” in the results section was revised as follows.

“[... omitted ...] Subsequently, by fitting to minimize the χ^2 values between theoretical and experimental difference curves, we selected the 10 best structures that best matched SADS₁ (or SADS₂ or SADS₃) (Supplementary Fig. 10). For each light-induced state (I₁, I₂, or P), the optimal structure that yielded the lowest χ^2 value for the experimental data (SADS₁, SADS₂ or SADS₃) among the 10 best structures was used to illustrate the representative structural characteristics of the protein in each state. (details in Methods and Supplementary Fig. 10). [... omitted ...]”

The following paragraph was added in the discussion section.

“For the light-induced dimerization, the structural landscapes also suggest that the light-induced dimerization of *AsLOV2* involves a gradual increase in the structural flexibility of the protein. L2 and L3 reveal gradual increases in the fluctuations of D1, D2, and the RMSD of the β-scaffold, indicating increasing structural flexibility of the Jα and A'α helices, as well as the β-scaffold, as the reaction proceeds. Notably, the fluctuation in the RMSD of the β-scaffold was smaller than those of D1 and D2, suggesting that the flexibility of the helices increases significantly more than that of the β-scaffold. Regarding this, the comparison of the 10 best structures of P shows differences in the overall structure, with significant structural variations in the Jα and A'α helices (Supplementary Fig. 10). Considering that the helices are located on the surface of *AsLOV2*, it is suggested that the global structure of *AsLOV2* would be flexible in the P state due to the structural flexibility of the helices. This global structural flexibility is

also observed by L1, which shows the gradual increases in the fluctuations of R_g and D_{max} (Fig. 5a).”

"MD simulations-aided structural analysis of TRXL data" in the method section was also revised as follows.

“[...] Based on the possible structural frames, for each state, the 10 best-fitted theoretical difference curves with the lowest χ^2_{red} values were selected as the best solutions. The 10 best protein structures were then selected from the MD-generated structures, which were used to generate the 10 best-fitted theoretical curves. These 10 best structures were subsequently used to generate the structural landscapes (Fig. 5), providing detailed structural information about the light-induced transition of *AsLOV2*. Furthermore, for each light-induced state (I₁, I₂, or P), the optimal structure with the smallest χ^2_{red} value among the 10 best structures was used to illustrate the representative structural characteristics of the protein in each state (Figs. 3, 4, and Supplementary Fig. 10). [... omitted ...]”

Regarding the reviewer’s comment on the residual secondary structure of the A’ α and J α helices in the P state, Supplementary Figure 10 and R6 indicate that the 10 best structures for the P state include both those with and without residual secondary helical structures in A’ α and J α helices. This suggests that the structure with the residual secondary helical structures could be considered as one possible snapshot for the P state.

Supplementary Figure 10. Results from the MD-aided structural analysis based on the SADSs. a–c Comparison of SADS (black) with the optimal (red) and 9 best-fit (gray) curves: (a) I₁, (b) I₂ and (c) P. **d–f** Optimal structure and 9 best structures: (d) I₁, (e) I₂ and (f) P. J α and A' α helices are depicted in blue and red, respectively, while the remaining parts are displayed in gray. In each panel of (d–f), the optimal structure is highlighted, while the rest 9 best structures are shown semi-transparently.

Figure R6. Examples of structures from the 10 best dimer structures for P, with and without residual secondary helical structures in A'α and Jα helices. a Example of structure for P with the residual helical properties. **b** Example of structure for P without the residual helical properties. These two structures were selected from the 10 best structures of P obtained in our structural analysis.

4. Table 1 shows a broad overview of previous time-resolved experiments on unfolding of helices in LOV domains. Results from the transient grating technique by Terazima and coworkers are not included although helix unfolding has been addressed by this scattering technique with precise time constants for different phototropin and aureochrome LOV domains (Refs 20-26). What is the reason for not including these results in Table 1?

→ It seems that the reviewer might have misunderstood the contents in Table 1. Our aim with Table 1 was to compare structural changes of solely AsLOV2 in our study with those reported in other AsLOV2 spectroscopic studies. Therefore, we did not discuss the unfolding time constants of other LOV domains in Table 1. The studies referred to by the reviewer (Refs 20-26 in the previous version by Terazima and coworkers) did not conduct a study on AsLOV2.

5. Page 17. The finding that A'alpha and Jalpha helices must both unfold for dimerization is not new and should be supported by references to previous studies (Zayner et al. 2012 J Mol Biol 419 61; Takeda et al. 2013 J Phys Chem B 117 15606; Herman et al. Biochemistry 2015 54 1484), which used exactly the same approach of truncating the extending helices.

→ We appreciate the reviewer's thoughtful concern. In the revised manuscript, we added citations for the studies mentioned by the reviewer (Zayner, J. P. et al., *J. Mol. Biol.* **419**, 61-74 (2012); Nakasone, Y. et al., *Biophys. J.* **91**, 645-653 (2006); Herman, E. et al., *Biochemistry* **54**, 1484-1492 (2015)) corresponding to the results of dimerization due to truncating A'α or J'α helices, as follows. The parts with major changes are indicated in bold.

“Furthermore, for the interaction between the helices and β-scaffold, the SEC profiles from constructs with the Jα helix deleted (ΔJα), the A'α helix deleted (ΔA'α), and both Jα and A'α helices deleted (ΔJα/ΔA'α) show the elution volume for the dimer conformation even in the dark state (Supplementary Figs. 13d–f), **consistent with previous studies from several LOV domains with truncated A'α or Jα helices**^{18, 64, 67.}”

Minor comments:

- *Abstract. Please use Avena sativa instead of Avena Sativa.*

→ Following the reviewer's comment, we have revised 'Avena Sativa' to 'Avena sativa' in the abstract.

- *Page 18. The denomination of light state as 'excited state' is misleading, because the real excited (triplet) state decays within a few microseconds prior to the experimental time window.*

→ We thank the reviewer for spotting this, which is now modified 'excited state' to 'light state' in the revised version.

- *Page 20. If a LOV protein is mentioned in the discussion, it must be stated whether this refers to phot1 or phot 2, LOV1 or LOV2.*

→ We thank the reviewer for spotting this, which is now modified in the revised version, as follows. The parts with major changes are indicated in bold.

“In particular, the dimerization of phot-LOV proteins, such as the *Arabidopsis thaliana* **phot1-LOV2** (*Atphot1-LOV2*) and *Chlamydomonas reinhardtii* **phot-LOV1** (*Crphot-LOV1*) have been observed on a timescale of tens of milliseconds, which is similar to the dimerization constant of 10.6 ms observed in this TRXL study²³⁻²⁵.”

REVIEWERS' COMMENTS

Reviewer #1 (Remarks to the Author):

I am very happy with the revision and the details in the rebuttal letter, all my questions have been addressed in detail. This is an excellent study and very detailed manuscript. I very strongly recommend that this revision is accepted for publication.

Reviewer #2 (Remarks to the Author):

In the new section "Elimination of the solvent heating contribution from TRXL data" - the phrase "Using a well-established method..." should be substantiated with a citation.

Reviewer #3 (Remarks to the Author):

The manuscript has been carefully revised and all major concerns have been well addressed. The additional controls performed such as the Falpha unfolding considerably strengthen the conclusions. Moreover, further supporting SAXS experiments on a truncated construct have been conducted. Previous findings on the light response of LOV domains are now adequately recognized. Some recommendations for possible further improvements are listed below as minor points.

Abstract. Ultrafast spectroscopic methods could not resolve the decisive event of LOV domains because adduct formation takes place later at a few microseconds. The authors might replace 'ultrafast' by 'time-resolved'.

Figure 6. The dimerization process is concentration-dependent. Accordingly, the experimental concentration might be included in the figure or the caption.

Line 468. Please correct that phototropin does not contain a histidine kinase, but a serine / threonine kinase.

Supplementary Figure 10 shows a beautiful and accurate overview of the representative structures, which might replace the simplified representation in Figure 6.

Responses to the comments from Reviewer #2

In the new section "Elimination of the solvent heating contribution from TRXL data" - the phrase "Using a well-established method..." should be substantiated with a citation.

→ To address the reviewer's comment, we have added the following citation in the section "Elimination of the solvent heating contribution from TRXL data".

"Kim, T. W. et al. Combined probes of X-ray scattering and optical spectroscopy reveal how global conformational change is temporally and spatially linked to local structural perturbation in photoactive yellow protein. *Phys. Chem. Chem. Phys.* **18**, 8911-8919 (2016)."

Responses to the comments from Reviewer #3

The manuscript has been carefully revised and all major concerns have been well addressed. The additional controls performed such as the Falpha unfolding considerably strengthen the conclusions. Moreover, further supporting SAXS experiments on a truncated construct have been conducted. Previous findings on the light response of LOV domains are now adequately recognized. Some recommendations for possible further improvements are listed below as minor points.

Abstract. Ultrafast spectroscopic methods could not resolve the decisive event of LOV domains because adduct formation takes place later at a few microseconds. The authors might replace 'ultrafast' by 'time-resolved'.

→ We have replaced 'ultrafast' with 'time-resolved' as the reviewer recommended.

Figure 6. The dimerization process is concentration-dependent. Accordingly, the experimental concentration might be included in the figure or the caption.

→ According to the reviewer's comment, we have added the following sentence to the caption of Figure 6.

"For the TRXL measurements of WT and I427V samples, each sample (WT or I427V) was prepared at a concentration of 1.5 mM in 20 mM Tris pH 7.0, 200 mM NaCl buffer."

Line 468. Please correct that phototropin does not contain a histidine kinase, but a serine / threonine kinase.

→ This has been corrected.

Supplementary Figure 10 shows a beautiful and accurate overview of the representative structures, which might replace the simplified representation in Figure 6.

→ Regarding the reviewer's comment, we have revised Figure 6 as follows.

Fig. 6. Structural dynamics of the *AsLOV2* photocycle revealed by TRXL study. The photocycle of *AsLOV2* includes the G state, three intermediates (I₁, I₂, and P) and related time constants (WT: 682 μs and 10.6 ms, and I427V: 130 μs and 3.4 ms), as determined from the kinetic analysis of the scattering data. For the TRXL measurements of WT and I427V samples, each sample (WT or I427V) was prepared at a concentration of 1.5 mM in 20 mM Tris pH 7.0, 200 mM NaCl buffer. The optimal structure of G was determined from the structural analysis on the static solution scattering data, while the optimal structures of three intermediates were extracted through the structural analysis using the TRXL data. The optimal structures (I₁ and I₂) indicate that the structural changes within the A'α and Jα helices allow the exposure of the β-scaffold to the external environment. Subsequently, *AsLOV2* undergoes dimerization (P), utilizing the dimeric interface formed between their β-scaffolds. For each state, the optimal structure is depicted alongside the 9 best structures, with the optimal structure highlighted and the remaining structures shown semi-transparently. The A'α and Jα helices are marked in red and blue, respectively, and the remaining structures are marked in gray.